# DCT-SNN: Using DCT to Distribute Spatial Information over Time for Learning Low-Latency Spiking Neural Networks

## Abstract

Spiking Neural Networks (SNNs) offer a promising alternative to traditional deep learning frameworks, since they provide higher computational efficiency due to event-driven information processing. SNNs distribute the analog values of pixel intensities into binary spikes over time. However, the most widely used input coding schemes, such as Poisson based rate-coding, do not leverage the additional temporal learning capability of SNNs effectively. Moreover, these SNNs suffer from high inference latency which is a major bottleneck to their deployment. To overcome this, we propose a scalable time-based encoding scheme that utilizes the Discrete Cosine Transform (DCT) to reduce the number of timesteps required for inference. DCT decomposes an image into a weighted sum of sinusoidal basis images. At each time step, a single frequency base, taken in order and modulated by its corresponding DCT coefficient, is input to an accumulator that generates spikes upon crossing a threshold. We use the proposed scheme to learn DCT-SNN, a low-latency deep SNN with leaky-integrate-and-fire neurons, trained using surrogate gradient descent based backpropagation. We achieve top-1 accuracy of 89.94%, 68.3% and 52.43% on CIFAR-10, CIFAR-100 and TinyImageNet, respectively using VGG architectures. Notably, DCT-SNN performs inference with 2-14X reduced latency compared to other state-of-the-art SNNs, while achieving comparable accuracy to their standard deep learning counterparts. The dimension of the transform allows us to control the number of timesteps required for inference. Additionally, we can trade-off accuracy with latency in a principled manner by dropping the highest frequency components during inference.

## 1 Introduction

Deep Learning networks have tremendously improved state-of-the-art performance for many tasks such as object detection, classification and natural language processing (Krizhevsky et al., 2012; Hinton et al., 2012; Deng & Liu, 2018). However, such architectures are extremely energy-intensive (Li et al., 2016) and hence require custom architectures and training methodologies for edge deployment (Howard et al., 2017). To address this, Spiking Neural Networks (SNNs) have emerged as a promising alternative to traditional deep learning architectures (Maass, 1997; Roy et al., 2019). SNNs are bio-plausible networks inspired from the learning mechanisms observed in mammalian brains. They are analogous in structure to standard networks, but perform computation in the form of spikes instead of fully analog values, as done in standard networks. For the rest of this paper, we refer to standard networks as Analog Neural Networks (ANNs) to distinguish them from their spiking counterparts with digital (spiking) inputs.The input and the correspondingly generated activations in SNNs are all binary spikes and inference is performed by accumulating the spikes over time. This can be visualized as distributing the one step inference of ANNs into a multi-step, very sparse inference scheme in the SNN.

The primary source of energy efficiency of SNNs comes from the fact that very few neurons spike at any given timestep. This event driven computation and the replacement of every multiply–accumulate (MAC) operation in the ANN by an addition in SNN allows SNNs to infer with lesser energy. This energy benefit can be further enhanced using custom SNN implementations with architectural modifications (Ju et al., 2020). (Li et al., 2017 ) have released a spiking version of the CIFAR-10 dataset based on inputs from neuromorphic sensors. IBM has designed a non-commercial processor 'TrueNorth' (F. Akopyan et al., 2015 ), and Intel has designed its equivalent 'Loihi' (Davies et al., 2018 ), that can train and infer on SNNs, and Blouw et al. (2019 ) have shown

SNNs implemented on Loihi to be two orders of magnitude more efficient than an equivalent ANN running on GPU for keywork spotting. However, a major challenge still to be addressed is that the accumulation of spikes over timesteps results in a higher inference latency in SNNs. Energy efficiency at the cost of too high a latency would still hamper real-time deployment. Consequently, reduction of timesteps required for inference in SNNs is an active field of research. One of the factors that significantly affects the number of timesteps needed is the encoding scheme that converts pixels into spikes over the timesteps. Currently, the most common encoding scheme is Poisson spike generation (Rueckauer et al., 2017), where the spikes at the input are generated as a Poisson spike train, with the mean spiking rate proportional to the pixel intensity. This scheme does not encode anything meaningful in the temporal axis and each timestep is the same as any other. Moreover, networks trained using this scheme suffer from high inference latency (Rueckauer et al., 2017). Temporal coding schemes such as phase (Kim et al., 2018) or burst (Park et al., 2019) coding have been introduced to better encode temporal information into the spike trains, but they still incur high latency and require a large number of spikes for inference. Another related temporal method is time-to-first-spike (TTFS) coding (Zhang et al., 2019; Park et al., 2020). They limit the number of spikes per neuron but the high latency problem still persists. Relative timing of spikes to encode information has been used in Comsa et al. (2020), but the results are only reported for simple tasks like MNIST and its scalability to deeper architectures such as VGG and more complex datasets like CIFAR remains unclear.

In this paper, we propose a novel encoding scheme to convert pixels into spikes over time. The proposed scheme utilizes a block-wise matrix multiplication to decompose spatial information into a weighted sum of basis, and then reverses the transform to allow reconstruction of the input over multiple timesteps. These bases, taken one per timestep, modulated by the weights from the forward transform are then presented to the spike generating layer. The spike generator sums the contribution of all bases seen until the current timestep, as shown in Figure 1. Though any invertible matrix can be utilized as the transform, the ideal transform follows the properties of energy compaction and orthonormality of bases as outlined in Section 3.1. We motivate Discrete Cosine Transform (DCT) as the ideal choice, since it is data independent, with orthogonal bases ordered by their contribution to spectral energy. Each timestep gets the information corresponding to a single base, starting from the zero frequency component at the first timestep. Each subsequent step refines the input representation progressively. At the end of the cycle, the entire pixel value has passed through the spike generating neuron. Thus, this methodology successfully distributes the pixel value over all the timesteps in a meaningful manner. Choosing the appropriate dimensions of the transform provides a fine grained control on the number of timesteps used for inference. We use the proposed scheme to learn DCT-SNN, a spiking version of an ANN and show that it cuts down the timesteps needed to infer an image taken from CIFAR-10, CIFAR-100 and TinyImageNet datasets from 100 to 48, 125 to 48 and 250 to 48, respectively, while achieving comparable accuracy to the state-of-the-art Poisson encoded SNNs. Additionally, ordering the frequencies bases being input at each timestep provides a principled way of trading off accuracy for a reduced number of timesteps during inference, if desired, by dropping the least important (highest frequency) components.

To summarize, the main contributions of this work are as follows,

- A novel input encoding scheme for SNNs is introduced wherein each timestep of computation encodes distinct information, unlike other rate-encoding methods.

- The proposed encoding scheme is used to learn DCT-SNN, which is able to infer with 2-14X lower timesteps compared to other state-of-the-art SNNs, while achieving comparable accuracy.

- The proposed technique is, to the best of our knowledge, the first work that leverages frequency domain learning for SNNs on vision applications.

- To the best of our knowledge, this is the first work that orders timesteps by significance to reconstruction. This provides an option to trade-off accuracy for faster inference by trimming some of the later frequency components, which is non-trivial to perform in other SNNs.

## 2 RELATED WORKS

**Learning ANNs in the frequency domain.** Successful learning for vision tasks in the frequency domain has been demonstrated in ANNs in several works. These utilize the DCT coefficients directly available from JPEG compression method (Wallace, 1992) without performing the decompression

steps. Conventional CNNs were used with DCT coefficients as input for image classification in Ulicny & Dahyot (2017) and Rajesh et al. (2019). Ehrlich & Davis (2019) proposed a model conversion algorithm to apply pretrained spatial domain networks to JPEG images. Wavelet features are utilized in Williams & Li (2016) to train CNN-based classifiers. However, these methods suffer a small accuracy degradation compared to learning in spatial domain. DCT features were used effectively for large scale classification and instance segmentation tasks in Xu et al. (2020). Although such frequency domain approaches have proved fruitful in ANNs, it is unexplored in SNNs despite the conversion of spatial bases of the image to temporal bases in the frequency domain being intuitively related to distributing the analog pixel values in ANNs to spikes over time in SNNs. There exist three prominent line of works for training SNNs, namely using spike-timing-dependent plasticity rules (STDP), ANN-SNN conversion and training from scratch. While STDP-based local learning (Diehl & Cook, 2015; Xu et al., 2020) is more bio-plausible, scaling such algorithms beyond MNIST type of tasks has been challenging. So, the following discussion focuses mainly on conversion and backpropagation based works.

**ANN-SNN Conversion.** The most common approach of training rate-coded deep SNNs is to first train an ANN and then convert it to an SNN for finetuning. (Diehl et al., 2015; Sengupta et al., 2019; Cao et al., 2015). Usually, the ANNs are trained with some limitations to facilitate this, such as not using bias, batch-norm or average pooling layers, though some works are able to bypass these constraints (Rueckauer et al., 2017). To convert ANNs to SNNs successfully, it is critical to adjust the threshold of Integrate-and-Fire (IF) / Leaky IF (LIF) neurons properly. The authors in Sengupta et al. (2019) recommend computing the layerwise thresholds as the maximum pre-activation of the neurons. This results in high accuracy but incurs high inference latency (2000-2500 timesteps). Alternatively, Rueckauer et al. (2017) choose a certain percentile of the pre-activation distribution as the threshold, reducing inference latency and improving robustness. The difference between these works and ours lie in the significance we attach to the timesteps.

**Backpropagation from Scratch and Hybrid Training.** Another approach to training SNNs is learning from scratch using backpropagation, which is challenging due to the non-differentiability of the spike function at the time of spike. Surrogate gradient based optimization (Neftci et al., 2019) has been utilized to circumvent this issue and implement backpropagation in SNNs effectively (Lee et al., 2020; Huh & Sejnowski, 2018). Surrogate gradient based backpropagation on the membrane potential at only a single timestep was proposed in Zenke & Ganguli (2018). Shrestha & Orchard (2018) compute the gradients using the difference between the membrane potential and the threshold, but only demonstrate on MNIST using shallow architectures. Wu et al. (2018) perform backpropagation through time (BPTT) on SNNs with a surrogate gradient defined on the membrane potential as it is continuous-valued. Overall, SNNs trained with BPTT using such surrogate-gradients have been shown to achieve high accuracy and low latency ($\sim$100-125 timesteps), but the training is very compute intensive compared to conversion techniques. Rathi et al. (2020) propose a combination of both methods, where a pre-trained ANN serves as initialization for subsequent surrogate gradient learning in the SNN domain. This hybrid approach improves upon conversion by reducing latency and speeding up convergence. However, this is orthogonal to the encoding scheme and can be used to improve the performance of any rate-coded scheme. In this work, we adopt the hybrid training method to train the SNNs. The key distinction of our method lies in how the pixel values are encoded over time, which is described next.

## 3 ENCODING SCHEME

An ideal encoding scheme to convert pixel values into spikes over time should capture relevant information in the temporal statistics of the data. Additionally, the total spike activity over all the timesteps at the input neuron should correspond to the pixel intensity. Our encoding scheme deconstructs the image into a weighted sum of basis functions. We invert this transform to reconstruct the image over time steps. Each basis function, taken one per timestep and modulated by the weights from the deconstruction is input to an Integrate-and-Fire (IF) neuron, which accumulates the input over timesteps and fires when accumulation crosses its threshold.

### 3.1 A GENERIC 1-D TRANSFORM TO DISTRIBUTE PIXELS OVER TIME

**1-D transformation.** For simplicity, we first consider a one dimensional transform over the entire input pixel space. Let us consider a single D-dimensional image, $X \in \mathbb{R}^{1 \times D}$. We transform this image using a transform matrix $T$ into a new coordinate system, where $T \in \mathbb{R}^{D \times D}$. The transformed

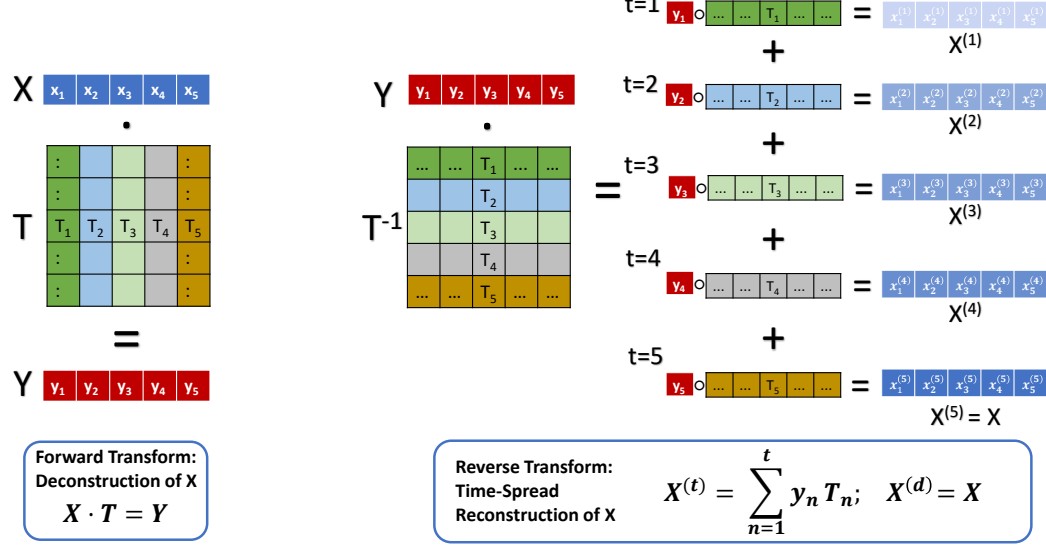

Figure 1: 1-D Encoding Scheme: On the left we show the **Forward Transform**. T represents the transform matrix that takes the input X into an intermediate coordinate system, resulting in representation Y. On the right, we show the **Inverse Transform** that uses Y to reconstruct X over time. Here $T^{-1} = T'$. The input image X is reconstructed progressively at each timestep by summing the basis vectors $T_n$ modulated by the corresponding coefficient $y_n$ over all previous timesteps. Since there are 5 bases shown here, X requires 5 timesteps for reconstruction.

vector, $Y = XT$, where $Y \in \mathbb{R}^{1 \times D}$, contains the coefficients of the image in the new coordinate system. This is shown pictorially for $d = 5$ in Figure 1. Assume that $T$ is a full rank matrix and let us consider $T^{-1} \in \mathbb{R}^{D \times D}$, the inverse transformation matrix that takes us back into the original coordinate system. For reasons clarified shortly, assume that T is an orthonormal matrix with its inverse equal to its transpose, $T^{-1} = T'$. The forward transform represents deconstruction of the input $X$ into a weighted sum of basis vectors, represented by the rows of $T^{-1}$, or columns of $T$ as shown in Figure 1. These bases are referenced by $T_n, n = 1, 2....d$. If we input one basis function per timestep to the SNN, we get intermediate representations of the input at timestep t, $X^{(t)}$ by modulating the $t$-th basis by its corresponding weight from the forward transform, summed over all previous timesteps. Summing over all bases allows us to reconstruct X. Mathematically,

$$X^{(t)} = \sum_{n=1}^{t} y_n T_n \qquad \text{and} \qquad X^{(d)} = X,$$

The analog value of $X^{(t)}$ is the input to the spike generator at each timestep and is converted to spikes using IF neurons as shown in Figure 2. Hence, we have successfully distributed the input X over d timesteps, with each timestep carrying information over our chosen bases. In the next section, we discuss the desirable properties of the bases for deconstructing X.

**Desirable Properties of the Basis Vectors.** The columns of the transform matrix $T$ contain the bases to deconstruct X. Since we use one basis per timestep, we want each base to offer non-interfering information about X. This is captured by the **orthonormality** constraint on T. Orthonormal columns avoid cancellation of information between timesteps, and relate the forward and reverse transforms by a transpose operation. The second constraint on T is that the bases be ranked by a measure of the information they carry. This allows each basis function to successively refine the representation per timestep. It is desirable to have the bulk of the information focused in the earlier timesteps, with fine-grained information added by the later steps. This **ordering of bases** allows us to drop bases in a principled manner to trade off accuracy for latency during inference.

**Transforms that Satisfy Constraints.** There are two widely used transforms that satisfy these properties: the **DCT** transform (Ahmed et al., 1974) and the Karhunen–Loève transform (Dony et al., 2001), also known as **Principal Component Analysis (PCA)**. DCT decomposes an image into a linear combination of sinusoidal frequencies, ranked by spectral energy. PCA uses the eigenvectors of the covariance matrix of the inputs as the bases, ranked by the amount of variance

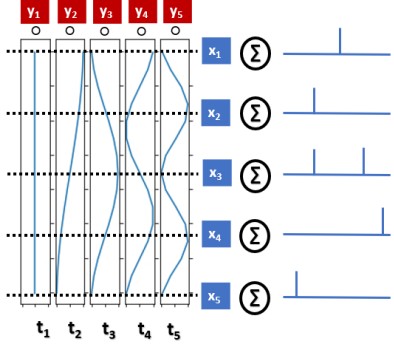

Figure 2: Spike generation with 1-D DCT basis functions input per timestep (shown vertically). The neuron spikes when the accumulated value crosses a threshold.

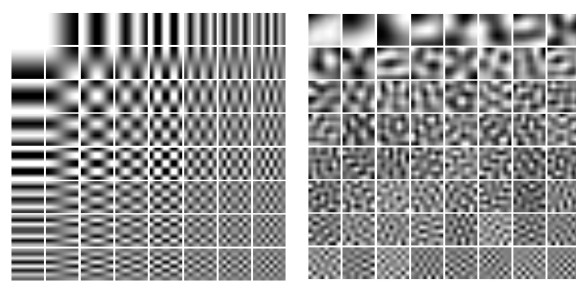

Figure 3: $8 \times 8$ 2-D DCT bases on the left and PCA bases for CIFAR-10 on the right. The DCT bases are ranked in a zig-zag fashion starting from top left to the bottom right and the PCA bases are ranked in significance from left to right and top to bottom.

they explain. DCT is commonly used in JPEG Compression and PCA in dimensionality reduction, by approximating the later components. However, PCA results in dataset dependent bases, whereas the DCT bases are pre-determined, avoiding extra computation. The 1-D DCT transform uses the following equation to take the pixel values $x_n$ into DCT coefficients $X_k$ using sinusoidal bases.

$$X_k = \sum_{n=0}^{N-1} x_n \cos\left[\frac{\pi}{N}\left(n + \frac{1}{2}\right)k\right] \qquad k = 0, \ldots, N-1. \tag{1}$$

Conversion of pixels denoted by $S_{xy}$ to DCT coefficients $F_{uv}$ for an $M \times N$ block is shown by:

$$C_x = \begin{cases} \frac{1}{\sqrt{2}} & \text{if } x = 0 \\ 1 & \text{else} \end{cases} \tag{2}$$

$$F_{uv} = \frac{2}{\sqrt{MN}} C_u C_v \sum_{x=0}^{M-1} \sum_{y=0}^{N-1} S_{xy} cos\left(u\pi\frac{2x+1}{2M}\right) cos\left(v\pi\frac{2y+1}{2N}\right) \tag{3}$$

The sinusoidal bases can be entered as the columns of a transformation matrix $T$. The forward transform is then computed as $Y = TXT'$ and the reverse transform is computed as $X = T'YT$. A comparison of the results for DCT, PCA, a random orthonormal transform without ranked bases, and a random non-orthonormal non-ranked transform is shown in Table 1. For the rest of the paper, we use the dataset agnostic DCT. Additionally, the conversion of spatial to temporal frequency in DCT lends itself intuitively to the concept of distributing spatial information in ANNs into spikes over time.

## 3.2 2-D Discrete Cosine Transform

We now extend the scheme to 2-D. The 2-D DCT is just the 1-D DCT applied first along the width channel and then along the length channel. Images are high dimensional, resulting in large transformation matrices $T$. This is undesirable since the number of DCT bases (or the dimension of $T$) dictates the number of timesteps required to reconstruct the image. To tackle this, similar to JPEG compression scheme, we first convert the image from RGB to YCbCr domain, and then take blocks of size $n \times n$ and perform 2-D DCT on these blocks, getting $n^2$ ordered frequency components. We replace the $n \times n$ pixel block with the equivalently reshaped frequency coefficients. An $n \times n$ block requires $n^2$ timesteps for perfect reconstruction of the pixel block. Small values of n allow us to reconstruct the images by summing over only a few basis images. In standard JPEG compression, $8 \times 8$ blocks are used, resulting in the 64 basis images shown in Figure 3. Similarly, the bases obtained from PCA on the training dataset of CIFAR-10 are also shown. Empirically, we find that block sizes of $4 \times 4$ converge to the best accuracy with the lowest number of timesteps. We usually need to run 3 full cycles to achieve convergence to best accuracy, amounting to $4 \times 4 \times 3 = 48$ timesteps. In each of the 3 cycles, we repeat the 16 DCT coefficients and bases, to allow time for spike propagation to the deeper layers. This is discussed in further detail in Section 4. Unlike the JPEG compression scheme, to improve accuracy we utilize an overlapped DCT scheme, where the

Table 1: Accuracy of VGG5 on CIFAR-10, with DCT block size = $4 \times 4$

| Transform Matrix | Accuracy (%) |
|---|---|
| Random | 64.7 |
| Unranked Orthonormal | 83.3 |
| PCA | 83.8 |
| DCT | 83.5 |

Table 2: Accuracy(%), with timesteps indicated in parenthesis. -p and -d represent training with pixels and DCT coefficients, respectively

| Configuration | VGG9 CIFAR-10 | VGG11 CIFAR-100 | VGG13 TinyImageNet |
|---|---|---|---|
| ANN-p | 91.3 | 69.7 | 56.9 |
| ANN-d | 90.4 | 66.4 | $45.5^a$ $53.1^b$ |
| SNN-p | 90.1 (175) 88.9 (100) | 67.8 (125) | 53 (250) |
| SNN-d | 88.2 (100) | 65.1 (125) | 44.6 (250) |
| DCT-SNN | 89.94 (48) | 68.3 (48) | 52.43 (125) 51.45 (48) |

[a] ANN without batchnorm and maxpool to facilitate conversion
[b] ANN with batchnorm and maxpool

blocks of pixels overlap. This is equivalent to performing convolution with a kernel size of 4 and a stride of 2, and increases our input dimensions by $4\times$. To counter this, we add an additional $2 \times 2$ average pooling layer before the linear layers.

### 3.3 Conversion from ANN and Threshold Selection for Spikes

In our scheme, the SNN trains on intermediate pixel representations, and hence we utilize an ANN trained with pixel intensities (rather than DCT coefficients) for initialization. The threshold of the IF neuron at the spike generator significantly affects the timesteps required for spike propagation to deeper layers. This IF neuron, as shown in Figure 2, receives the bases modulated by the DCT coefficients and accumulates them over timesteps, firing when the accumulation crosses the threshold. We allow for both positive and negative spikes to account for the positive and negative cycles of the sinusoidal bases. Similar to the hidden layer neurons, the threshold is chosen as a percentile of the accumulation at the spike generator neurons. We obtained best results by using 6.5 and 93.5 percentile of the accumulation as thresholds for negative and positive spikes, respectively.

### 4 Experiments and Results

We implement DCT-SNN by incorporating the proposed encoding scheme with surrogate-gradient based learning using LIF neurons. Starting with a pretrained ANN, we copy the weights to the SNN and select the 99.9 percentile of the pre-activation distribution at each layer as its threshold. The details of the learning methodology and hyperparameters of training are given in appendix section A.1 and A.2. The implementation is provided as part of the supplementary material.

**Choice of Transformation.** We first analyze the performance of DCT-SNN trained on different choices of transformation matrices (denoted as $T$ in section 3.1). Table 1 shows the results for a VGG5 network trained on CIFAR-10. With a random $T$, the network does converge but with much lower accuracy than the ANN. Next, to avoid interference between different bases, we use a random orthonormal $T$. Table 1 shows that the accuracy improves by $\sim 20\%$ compared to non-orthonormal case. However, this choice of $T$ does not perform energy compaction. Ranking bases by their contribution to reconstruction allows us to trade off accuracy for latency during inference. To incorporate this, we experiment with the transformation matrix generated by performing PCA on $4 \times 4$ blocks of the inputs from the training dataset. While this satisfies both the desired properties and gives the best performance, it is a data-dependent transform. Therefore, we utilize the fixed DCT matrix, and find that it performs at par with PCA, while additionally being data-agnostic. For all subsequent analysis, we use DCT as the choice of transformation.

**Effect of Block Size and Overlap.** Having chosen DCT to determine the bases of our encoding scheme, we tune the block size and stride. The results are shown in Fig. 4. 'DCT-x' denotes a network trained on inputs transformed with DCT of blocksize $x$, and 'ov' refers to overlap among the DCT blocks. Reducing the blocksize from 16 to 4 improves accuracy consistently. Moreover, since a blocksize of $x$ requires $x^2$ timesteps to pass one information of cycle, smaller blocksizes benefit from a lesser requirement of timesteps per cycle. The results on different block sizes with

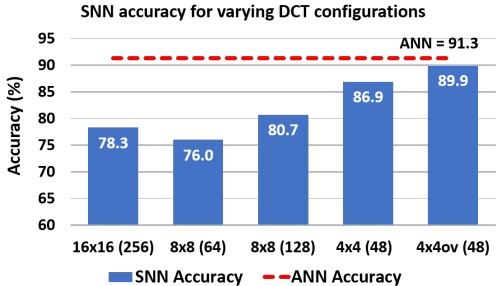

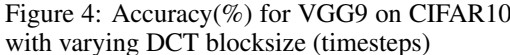

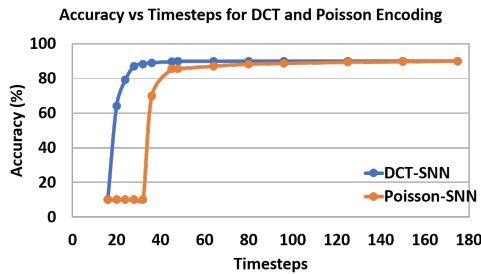

Figure 4: Accuracy(%) for VGG9 on CIFAR10 with varying DCT blocksize (timesteps)

Figure 5: Accuracy(%) for VGG9 on CIFAR10 with varying timesteps

timesteps required in parenthesis are shown in Fig. 4. We empirically find that DCT-2 is unable to converge, and that overlapped version of DCT-4 with a stride of 2 outperforms all other cases. Hence, we utilize this scheme for all further experiments.

**Number of Cycles for Information Propagation**. The next design parameter is the number of timesteps per forward pass. The performance of DCT-SNN trained with different timesteps is shown in Fig. 5. In the scheme DCT-4, one full cycle amounts to 16 timesteps. The network converges to 89.94% accuracy with 48 timesteps and 88.41% accuracy with just 32 timesteps. Since performance saturates after 3 cycles (48 timesteps), we choose 48 as the number of timesteps to train DCT-SNN on CIFAR-10 and CIFAR-100. However, for deeper networks and larger datasets, larger timesteps might yield further improvements, as shown in Table 2 for TinyImagenet with VGG13. Notably, the accuracy of Poisson-encoded networks drops severely below 45 timesteps (Fig. 5), whereas DCT-SNN suffers a minimal drop even with 28 timesteps. In particular, Poisson does not converge for 32 timesteps or lesser, whereas we achieve less than 2% accuracy drop at 32 timesteps.

**Results on CIFAR and TinyImageNet.** The experimental results using the proposed scheme for CIFAR and TinyImageNet datasets are shown in Table 2. DCT-SNN performs comparably to SNNs trained with Poisson-encoded pixels, but requires lesser than half the timesteps. We also compare our results with SNNs trained on Poisson-encoded DCT coefficients and show that our method performs better, presumably due to the reconstruction of pixels over time. To demonstrate the scalability of the proposed algorithm, we apply it to the TinyImagenet dataset. SNNs with Poisson-encoded pixels require $\sim 250$ timesteps to converge, whereas our method can converge to comparable accuracy in 125 timesteps. Allowing for a 1% drop in accuracy, our method converges with even 48 timesteps.

**Performance Comparison.** We collect reported results for different state-of-the-art SNNs and compare our performance in Table 3. DCT-SNN performs better than or comparably to the reported accuracy of the these methods, while achieving lower latency for inference. Wu et al. (2018) report CIFAR10 results with 30 timesteps for a shallow network with 2 convolutional and 2 fully-connected layers with 50.7% accuracy. We implement the same net with DCT-SNN and achieved 68.1% accuracy with 28 timesteps. Next, we compare with methods that expose analog pixel intensities directly to the first convolutional layer, instead of spikes. In a subsequent work, Wu et al. (2019b) achieve 90.53% accuracy on CIFAR-10 using just 12 timesteps on a network with 5 convolutional and 2 fully connected layers. After each conv layer, the binary activations go through a channel-wise normalization (termed as neunorm in the paper). This makes the binary activations essentially analog, as can be seen from Eqn. (9) and (10) in Wu et al. (2019b). It is unclear whether the efficiency of 12 timesteps is arising due to their encoding scheme, because of the proposed channel-wise normalization resulting in analog computation (MAC instead of just accumulation) at each layer, or their voting scheme based on class-wise populations. We believe that the analog nature of computation at each layer makes this network closer to ANNs than SNNs, resulting in the significant reduction in timesteps, especially since their network is shown to converge to a good accuracy in a single timestep.

We also compare with two works that train ANNs on DCT coefficients. Ehrlich & Davis (2019) report ANNs with 72.5% and 38.5% accuracy on CIFAR-10 and CIFAR-100, respectively and Ulicny & Dahyot (2017) report 86.35% accuracy on CIFAR-10. DCT-SNN reaches upto $\sim 90\%$ accuracy for CIFAR-10, as seen in Table 2, achieved by performing DCT on blocks of size $4 \times 4$ with an overlap of 2, unlike the standard JPEG scheme of $8 \times 8$ with no overlap. However, this does not train in the frequency domain unlike Ehrlich & Davis (2019), since after passing the modulated

Table 3: Comparison of DCT-SNN to other reported results. SGB denotes Surrogate-Gradient Based backprop, Hybrid denotes pretrained ANN followed by SNN fine-tuning, TTFS denotes Time-To-First-Spike scheme, TL denotes tandem learning and (xC, yL) denotes an architecture with x Conv layers and y Linear layers.

| Reference | Dataset | Training | Architecture | Accuracy(%) | Timesteps |
|---|---|---|---|---|---|
| (Hunsberger & Eliasmith, 2015) | CIFAR10 | Conversion | 2C, 2L | 82.95 | 6000 |
| (Cao et al., 2015) | CIFAR10 | Conversion | 3C, 2L | 77.43 | 400 |
| (Sengupta et al., 2019) | CIFAR10 | Conversion | VGG16 | 91.55 | 2500 |
| (Lee et al., 2020) | CIFAR10 | SGB | VGG9 | 90.45 | 100 |
| (Rueckauer et al., 2017) | CIFAR10 | Conversion | 4C, 2L | 90.85 | 400 |
| (Rathi et al., 2020) | CIFAR10 | Hybrid | VGG9 | 90.5 | 100 |
| (Park et al., 2020) | CIFAR10 | TTFS | VGG16 | 91.4 | 680 |
| (Park et al., 2019) | CIFAR10 | Burst-coding | VGG16 | 91.4 | 1125 |
| (Kim et al., 2018) | CIFAR10 | Phase-coding | VGG16 | 91.2 | 1500 |
| (Wu et al., 2018) | CIFAR10 | SGB | 2C, 2L | 50.7 | 30 |
| (Wu et al., 2019b) | CIFAR10 | SGB | 5C, 2L | 90.53 | 12 |
| (Wu et al., 2019a) | CIFAR10 | TL(LIF) | 5C, 2L | 89.04 | 8 |
| **This work** | **CIFAR10** | **DCT-SNN** | **VGG9** | **89.94** | **48** |
| (Lu & Sengupta, 2020) | CIFAR100 | Conversion | VGG15 | 63.2 | 62 |
| (Rathi et al., 2020) | CIFAR100 | Hybrid | VGG11 | 67.9 | 125 |
| (Park et al., 2020) | CIFAR100 | TTFS | VGG16 | 68.8 | 680 |
| (Park et al., 2019) | CIFAR100 | Burst-coding | VGG16 | 68.77 | 3100 |
| (Kim et al., 2018) | CIFAR100 | Phase-coding | VGG16 | 68.6 | 8950 |
| **This work** | **CIFAR100** | **DCT-SNN** | **VGG11** | **68.3** | **48** |
| **This work** | **TinyImagenet** | **DCT-SNN** | **VGG13** | **52.43** | **125** |

bases through the network, we have passed the equivalent information of the input image in the pixel domain. To verify that our method is trainable via backpropagation from scratch, we trained a VGG9 SNN from scratch for CIFAR-10, which gives $84.9\%$ accuracy with 48 timesteps. We show a more detailed comparison with other encoding schemes in Appendix section A.5.

**Accuracy-Latency Trade-off.** The ranking of bases in our scheme allows us to drop the least significant components. In Figure 6, we show the effect the ranking of bases has on accuracy by performing inference on VGG9 DCT-SNN trained on CIFAR-10 for 48 timesteps using all 16 frequencies. A minimum of 16 timesteps (1 cycle) are required for spike propagation to the deeper layers, and hence any configuration with timesteps lesser than 16 cannot infer correctly. We provide 2 cycles of inputs on the test data. The first cycle uses all 16 components, and the next adds successively higher frequencies. Due to the fact that the bulk of the information is contained in the earlier timesteps, we are able to get good accuracy (73.9% out of 88.6% ) with just the first 4 bases. Successive components add more refined information, and therefore the accuracy saturates, as evident from Figure 6. To the best of our knowledge, this is the first work that demonstrates a principled tradeoff between inference accuracy and latency on a trained network. Results for networks trained with limited frequencies are shown in Appendix A.3. The effect of changing the order of inputting frequencies is shown in Appendix section A.6.

**Computational Efficiency.** The floating-point (FP) MAC operations in ANN are replaced by FP additions in SNN. The cost of a MAC operation ($4.6pJ$) is $5.1\times$ compared to an addition ($0.9pJ$) (Horowitz, 2014) in 45nm CMOS technology. The expressions representing the computational cost in the form of operations per layer in an ANN, #ANN$_{\text{ops}}$, are given in Appendix A.4. The number of operations per layer in an equivalent DCT-SNN are related to #ANN$_{\text{ops}}$ by the layer's spike-rate.

$$\#\text{DCT-SNN}_{\text{ops, L}} = \text{spike rate}_{\text{L}} \times \#\text{ANN}_{\text{ops, L}},$$

where spike rate$_{\text{L}}$ is the average number of spikes per neuron per image over all timesteps in layer L. The layerwise spike rates for CIFAR-10 and CIFAR-100 using DCT-SNN are shown in Fig. 7. The overall average spike rate considering all layers for both cases is well below 5.1 (relative cost of MAC to addition), indicating the energy benefits of DCT-SNN over the corresponding ANN. For the DCT-SNN, there are additional MAC operations for 2 full precision matrix multiplications in

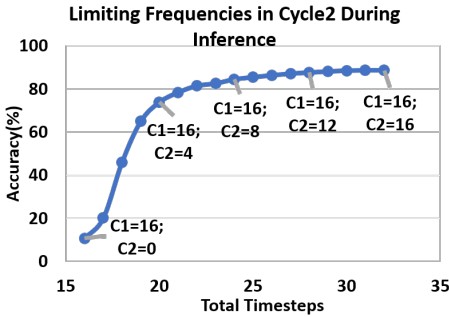

Figure 6: Accuracy-latency tradeoff during inference; VGG9 trained on CIFAR-10 with all 16 frequencies for 48 timesteps. During inference, cycle1 uses all 16 frequencies, cycle2 uses limited, ordered frequencies.

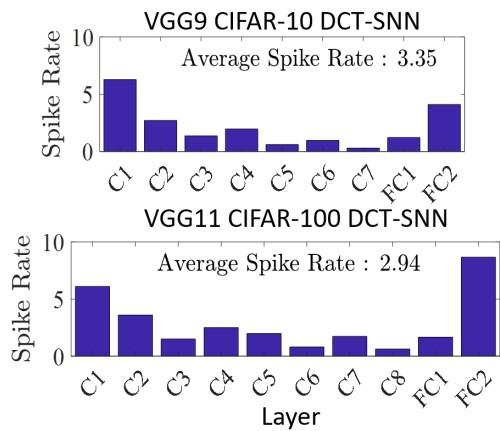

Figure 7: DCT-SNN layerwise spike rate. C and FC denote Conv and Fully Connected layers.

the preprocessing step (the forward and reverse transforms). We denote these as Encoder$_{ops}$. This computation is needed for only one cycle (16 timesteps), since the other 2 cycles just repeat the same bases and coefficients. The overhead is negligible when compared to the number of operations over all the layers across all timesteps. We compute the energy benefits of DCT-SNN over ANN, $\alpha$ as ,

$$\alpha = \frac{E_{ANN}}{E_{DCT\text{-}SNN}} = \frac{\sum_L \#ANN_{ops,L} * 4.6}{\#Encoder_{ops} * 4.6 + \sum_L \#DCT\text{-}SNN_{ops,L} * 0.9}. \tag{4}$$

The values of $\alpha$ are 1.52 and 1.74 for VGG9-CIFAR10 and VGG11-CIFAR100, respectively, demonstrating that DCT-SNN provides improved energy efficiency over its ANN counterpart. Similar to Park et al. (2020), the cost of memory access has not been considered in this evaluation, since it depends on the hardware architecture and system configurations. This efficiency metric $\alpha$ can be further enhanced by reducing the encoder overhead by approximating the DCT transformation with integer transform (IT). IT can be performed with just shift and add operations. We implemented IT-based encoding for a VGG9 SNN on CIFAR-10 and obtained 89.2% accuracy with 48 timesteps.

## 5 CONCLUSION

Bio-plausible SNNs offer a promising energy-efficient alternative to ANNs. The inputs to SNNs are available as spikes over time, instead of analog pixel values. The SNN derives efficiency from the sparsity of spikes per timestep, combined with their event-driven computation. However, the spike distribution over timesteps causes SNNs to have a high inference latency. The most widely used Poisson based rate encoding scheme does not encode any meaningful information into the temporal axis of SNNs, and requires a large number of timesteps for inference. To address this, we propose a new encoding scheme that can be used to distribute spatial pixel information over timesteps in an ordered fashion. The proposed scheme utilizes a linear transform in the form of an invertible matrix, with columns that serve as basis of representation distribution. The input pixels are reconstructed over time by summing these bases modulated by the coefficients from the intermediate representation. At each step, we feed in the product of one of these bases and the corresponding DCT coefficient, by order of their contribution, to an integrate-and-fire neuron at the input layer. As we cycle through all bases, the neuron receives the total pixel value spread over all timesteps. The ideal properties of the bases are orthonormality to avoid interference with each other, and ordering by their contribution to reconstruction of the pixel intensities. The choice of the transformation matrix are thus PCA and DCT. We chose DCT due to the frequency bases being dataset-agnostic. We get best performance with 2-D DCT on $4 \times 4$ blocks of the input, resulting in 16 basis frequencies. We show that passing these 16 bases through the SNN cyclically a few times gives DCT-SNN comparable accuracy to their ANN counterparts, with less than half the number of timesteps required to learn other state-of-the-art SNNs. Additionally, ranking these bases differentiates one timestep from another. Consequently, we are able to drop the least important bases (and therefore, timesteps) and still infer, albeit with lower accuracy. This principled trade-off between inference accuracy and latency is a promising direction for deploying SNNs on edge devices.

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

## A    APPENDIX

### A.1    OVERALL TRAINING METHODOLOGY

#### A.1.1    SPIKING NEURON MODEL

In this work, we employ the bio-plausible Leaky Integrate and Fire (LIF) model (Hunsberger & Eliasmith, 2015), which is described by-

$$\tau_m \frac{dU}{dt} = -(U - U_{rest}) + RI, \quad U \leq V_{th} \tag{5}$$

where $U$ denotes the membrane potential, $I$ is the input current representing the weighted summation of spike-inputs, $\tau_m$ indicates the time constant for membrane potential decay, $R$ represents membrane leakage path resistance, $V_{th}$ is the firing threshold and $U_{rest}$ is resting potential. The discretized version of Eqn. 5 implemented in our work is given as-

$$u_i^t = \lambda u_i^{t-1} + \sum_j w_{ij} o_j^t - v_{th} o_i^{t-1}, \tag{6}$$

$$o_i^{t-1} = \left\{ \begin{array}{ll} 1, & if \; u_i^{t-1} > v_{th} \\ 0, & otherwise \end{array} \right. \tag{7}$$

where $u$ is the membrane potential, subscripts $i$ and $j$ represent the post and pre-neuron, respectively, t denotes timestep, $\lambda$ is the leak constant= $e^{\frac{-1}{\tau_m}}$, $w_{ij}$ represents the weight of connection between the i-th and j-th neuron, $o$ is the output spike, and $v_{th}$ is the firing threshold. As evident from Eqn. 6, whenever $u$ crosses this threshold, it is reduced by the amount $v_{th}$, implementing a soft-reset. We implement the proposed DCT-SNN using the model described above with our encoding scheme, the code for our work is submitted as part of the supplementary material.

#### A.1.2    SURROGATE-GRADIENT BASED LEARNING

To train deep SNNs, we use surrogate-gradient based backpropagation which performs both the temporal as well as the spatial credit assignment of errors. Spatial credit assignment is achieved by spatial error distribution across all layers, while for temporal credit assignment, we unroll the network in time and employ backpropagation through time (BPTT) (Werbos, 1990). The output layer neuronal dynamics is given as-

$$u_i^t = u_i^{t-1} + \sum_j w_{ij} o_j^t, \tag{8}$$

here the $u_i$s correspond to the membrane potential of i-th neuron of final (L-th) layer. The final layer neurons do not spike, rather we just accumulate their potential over time for classification purpose. These accumulated outputs are passed through a softmax layer to obtain the class-wise probability distribution and then the cross-entropy between the true output and the network's predicted distribution is used as loss for backpropagation. The governing equations are-

$$L = -\sum_i y_i log(z_i), \tag{9}$$

$$z_i = \frac{e^{u_i^T}}{\sum_{k=1}^N e^{u_k^T}}, \tag{10}$$

where L is the loss function, y denotes true output, z is the prediction, T is the total number of timesteps and N is the number of classes. The derivative of the loss w.r.t. to the membrane potential of the neurons in the final layer is given as-

$$\frac{\partial L}{\partial u_i^T} = z_i - y_i, \tag{11}$$

and the weight updates at the output layer are done as-

$$w_{ij,L} = w_{ij,L} - \eta \Delta w_{ij,L}, \tag{12}$$

$$\Delta w_{ij,L} = \sum_t \frac{\partial L}{\partial w_{ij,L}^t} = \sum_t \frac{\partial L}{\partial u_i^T} \frac{\partial u_i^T}{\partial w_{ij,L}^t} = \frac{\partial L}{\partial u_i^T} \sum_t \frac{\partial u_i^T}{\partial w_{ij,L}^t}, \tag{13}$$

where $\eta$ is the learning rate, and $w_{ij,L}^t$ is the weight between i-th neuron at layer $L$ and j-th neuron at layer $L-1$ at timestep t. Since the output layer neurons are non-spiking, the non-differentiability is not an issue here. On the other hand, the hidden layer parameter update is given by-

$$\Delta w_{ij,k} = \sum_t \frac{\partial L}{\partial w_{ij,k}^t} = \sum_t \frac{\partial L}{\partial o_{i,k}^t} \frac{\partial o_{i,k}^t}{\partial u_{i,k}^t} \frac{\partial u_{i,k}^t}{\partial w_{ij,k}^t}, \quad k = 2, 3, ...L-1 \tag{14}$$

where $o_{i,k}^t$ is the spike-generating function (Eqn. 7), k is layer index. We approximate the gradient of this function w.r.t. its input using the linear surrogate-gradient (Bellec et al., 2018) as-

$$\frac{\partial o}{\partial u} = \gamma max\{0, 1 - |\frac{u - v_{th}}{v_{th}}|\}, \tag{15}$$

where $\gamma$ is a hyperparameter chosen as 0.3 in this work.

---

**Algorithm 1** Procedure of spike-based backpropagation learning for an iteration.

---

**Input:** pixel-based mini-batch of input $(X)$ - target $(Y)$ pairs, total number of timesteps $(T)$, number of layers $(L)$, pre-trained ANN weights $(W)$, membrane potential $(U)$, membrane leak constant $(\lambda)$, array of layer-wise firing thresholds $(V_{th})$, dct block-size $b$, number of freq. component to train with $f$
**Initialize:** $U_l^t = 0$, $\forall l = 1, ..., L$
// $M$= generate dct encoded-inputs for the current mini-batch data for $b * b$ timesteps
// **Forward Phase**
**for** $t \leftarrow 1$ **to** $T$ **do**
   $O_1^t = M[t\%f]$; // here M[x] denotes the dct-encoded inputs sampled from frequency index x
   **for** $l \leftarrow 2$ **to** $L-1$ **do**
      // membrane potential integrates weighted sum of spike-inputs
      $U_l^t = U_l^{t-1} + W_l * O_{l-1}^t$
      **if** $U_l^t > V_{th}$ **then**
         // if membrane potential exceeds $V_{th}$, a neuron fires a spike
         $O_l^t = 1$, $U_l^t = U_l^t - V_{th}$
      **else**
         // else, membrane potential decays exponentially
         $O_l^t = 0$, $U_l^t = \lambda * U_l^t$
      **end if**
   **end for**
   // final layer neurons do not fire
   $U_L^t = U_L^{t-1} + W_L * O_{L-1}^t$
**end for**
//calculate loss, Loss=cross-entropy$(U_L^T, Y)$
// **Backward Phase**
**for** $t \leftarrow T$ **to** $1$ **do**
   **for** $l \leftarrow L-1$ **to** $1$ **do**
      // evaluate partial derivatives of loss with respect to weight by unrolling the network over time
      $\triangle W_l^t = \frac{\partial Loss}{\partial O_l^t} \frac{\partial O_l^t}{\partial U_l^t} \frac{\partial U_l^t}{\partial W_l^t}$
   **end for**
**end for**

---

### A.1.3 WEIGHT INITIALIZATION AND THRESHOLD BALANCING

A key component in successful implementation of SNNs is proper initialization of weights and thresholds. As mentioned in section 2 of the main manuscript, we first pre-train an analogous ANN and copy the weights to the SNN for finetuning. It is critical to balance the layerwise neuronal thresholds to achieve satisfactory performance in SNNs. One approach is to choose the maximum

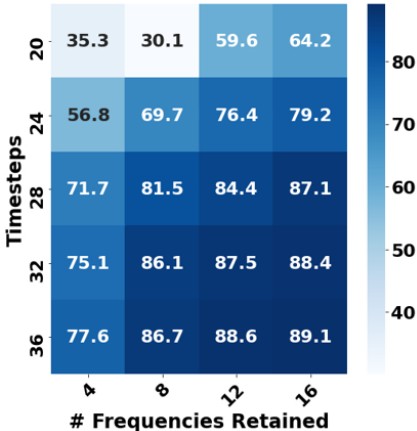

Figure 8: Training Accuracy with Limited Frequencies.

input to the neurons computed over all timesteps at each layer as the threshold at that corresponding layer (Sengupta et al., 2019). However, empirically, we have found this scheme to be unstable (training did not converge in some cases due to spike-vanishing in the deeper layers), hence we select the 99.9 percentile of the pre-activation distribution at each layer to be that layer's threshold. Again, such threshold balancing has been argued to be more robust (Rueckauer et al., 2017). Notably, the threshold computation has to be performed one layer at a time and sequentially from first layer to the end. Having initialized the SNN as discussed above, we perform the surrogate-gradient based learning, the details of which is depicted in Algorithm 1. In addition, next we also provide the experimental details in appendix section A.2.

## A.2 EXPERIMENTAL DETAILS

### A.2.1 DATASETS AND MODELS

We perform our experiments on VGG9 for CIFAR10 dataset, VGG11 for CIFAR100 and VGG13 for TinyImagenet. Some comparisons with other encoding schemes are done using VGG5.

### A.2.2 TRAINING PARAMETERS

For all datasets, we follow some standard data augmentation techniques such as padding by 4 pixels on each side, and a $32 \times 32$ crop is randomly sampled from the padded image or its horizontally flipped version (with 0.5 probability of flipping). While testing, the original $32 \times 32$ images are used. Both training and testing data are normalized using 0.5 as mean and standard deviation for all channels. For training the ANNs, we use cross-entropy loss with stochastic gradient descent optimization (weight decay=0.0001, mometum=0.9). In the ANN domain, VGG5, VGG9 and VGG11 are trained for a total of 300 epochs, with an initial learning rate of 0.1, which is divided by 10 at each 100-th epoch. VGG13 with TinyImagenet is trained with similar learning rate schedule, but with initial learning rate of 0.01. The ANNs are trained with some architectural constraints to avoid significant loss during subsequent ANN-SNN conversion (Diehl et al., 2015; Sengupta et al., 2019). The ANNs do not have bias terms as it increases the difficulty of threshold balancing. Again, batch-normalization is not used, rather dropout (Srivastava et al., 2014) is used as the regularizer and a constant dropout mask is used across all timesteps while training in SNN domain. Furthermore, average-pooling is used to reduce the feature map size since max-pooling causes significant information loss in SNNs (Diehl et al., 2015). During training the ANN, the weights are initialized using Xavier initialization (Glorot & Bengio, 2010). After conversion, for training in the SNN domain, networks are trained for 20-30 epochs with cross-entropy loss and adam optimizer (weight decay=0.0005). Initial learning rate is kept at 0.0001, which is halved every 5 or 6 epochs. The leak constant $\lambda$ is chosen as 0.9901 for all simulations.

### A.3 TRAINING WITH CURTAILED FREQUENCIES

We show the effect of training with limited frequencies in Figure 8. The frequencies are repeated cyclically until the specified number of timesteps. For instance, the accuracy for 8 frequencies given 3 times each (24 timesteps) is 69.7%. We note that during training, there is no benefit to dropping frequencies at iso-latency requirements.

### A.4 COMPUTATIONAL COST

The equations for calculating the number of operations in a particular layer of an ANN are given by

$$\#\text{ANN}_{\text{ops}} = \begin{cases} k_w \times k_h \times c_{in} \times h_{out} \times w_{out} \times c_{out}, & \text{Conv layer} \\ n_{in} \times n_{out}, & \text{Linear layer} \end{cases}$$

where $k_w(k_h)$ denote filter width (height), $c_{in}(c_{out})$ is number of input (output) channels, $h_{out}(w_{out})$ is the height (width) of the output feature map, and $n_{in}(n_{out})$ is the number of input (output) nodes.

### A.5 PERFORMANCE COMPARISON WITH DIFFERENT ENCODING SCHEMES

### A.5.1 PERFORMANCE COMPARISON WITH TEMPORAL ENCODING SCHEMES ON MNIST

Table 4: Accuracy of various temporal encoding schemes on MNIST

| Reference | Accuracy(%) | Timesteps |
|---|---|---|
| (Kheradpisheh & Masquelier, 2020 ) | 97.4 | 256 |
| (Comsa et al., 2020) | 97.9 | not reported |
| (Stephan et al., 2020) | 85 | 10 |
| (Yu et al., 2013) | 78 | 100 |
| (Xu et al., 2018) | 87 | not reported |
| (Beyeler et al., 2013) | 91.6 | 500 |
| This work | 98.54 | 16 |
| This work | 86.7 | 2 |
| This work | 97.3 | 5 |

To further compare the performance of the proposed DCT-SNN encoding scheme with recent temporal methods on MNIST, we implement it on a shallow network with just 1 hidden layer consisting of 784-100-10 neurons (all fully-connected). The results are reported in Table 4. As can be seen, our methods outperforms these recent temporal methods in terms of accuracy and also converges at much lower timesteps.

### A.5.2 PERFORMANCE COMPARISON WITH DIFFERENT ENCODING SCHEMES ON CIFAR

To further compare our results with other reported works in terms of timesteps required at iso-accuracy level, we depict the inference curve across different timesteps in Fig. 9. The figure is adopted from Fig. 6 of Park et al. (2020) and demonstrates the results of "T2FSNN" encoding scheme, which is a temporal encoding scheme and other rate and temporal encoding schemes such as "Rate" (Rueckauer et al., 2017), "Phase" (Kim et al., 2018), and "Burst" coding (Park et al., 2019). The left graph in Fig. 9 is recreated for CIFAR-10, and shows $\sim 200$ timesteps for the fastest convergence among these encoding methods, in contrast, we achieve $\sim 90\%$ accuracy in just 48 timesteps, saturating far earlier than any of these methods. From Fig. 6 of Park et al. (2020), we can tell that the best version of "T2FSNN" first reaches 90% roughly at 240 steps, "Burst" at 300, "Phase" at 425, and "Rate" at 1200 timesteps, showing that we reduce latency by orders of magnitude, resulting in convergence at much fewer timesteps. The network is slightly different here,

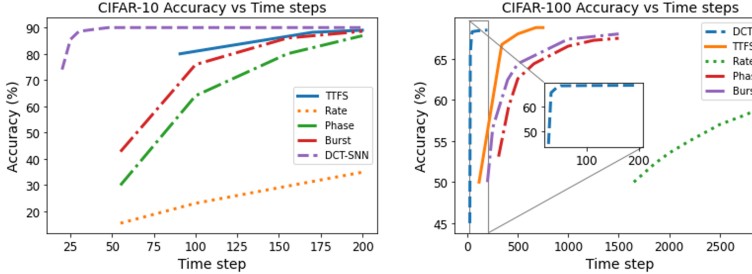

Figure 9: Accuracy versus Latency curve for various coding schemes, the values for TTFS (Park et al., 2020), Phase (Kim et al., 2018), Burst (Park et al., 2019) and rate (Rueckauer et al., 2017) have been adopted from Park et al. (2020).

.

ours is VGG9 and the network used in Park et al. (2020) is VGG16, but in our opinion, that affects final convergence accuracy more than it affects orders of magnitude of timesteps for inference. Similarly, we exceed 68% accuracy in 48 timesteps when training a VGG11 on CIFAR100. The graph on the right in Fig. 9 shows the convergence statistics for VGG16 on CIFAR100 using "T2FSNN", "Burst", "Phase" and "Rate". The best version of "T2FSNN" reaches 68% roughly at 500 steps, "Burst" at 1500, "Phase" at 2000, and "Rate" does not go above 60% in even 3000 timesteps.

## A.6 TRAINING WITH INTERLEAVED AND INTERMITTENT FREQUENCIES

In this section, we analyze the effect of training with (a) interleaved and (b) intermittent frequencies, instead of all 16 frequency components given in a cyclic order.

For the interleaved case, instead of giving input as frequencies 0,1,2,...15,0,1,2,....15,0,1,2,...15 we input them as 0,0,0,1,1,1,2,2,2, ....15,15,15. So, one specific basis is repeated for 3 subsequent timesteps before giving the next frequency as input. Our original cyclic scheme gave the best reported accuracy of 89.94% and the interleaved scheme only achieved 79.7%. A similar cyclic vs interleaving experiment was done with the frequencies limited to top 8, repeated for 3 cycles (24 timesteps). The cyclic scheme achieved 69.7% and the interleaved achieved 53.73%. We believe this drop is due to the resetting of membrane potential as it fires between timesteps, causing temporal dependency to be incorporated between different timesteps and interleaving cannot leverage this dependency. Additionally, since the earlier DCT coefficients contain most of the energy, the spikes at the later timesteps start dying out with interleaved frequencies.

Next, we tried giving intermittent frequencies such as (0,2,5,7,9,10,12,15) given cyclically for 6 cycles (48 timesteps) and got 84.4%, an expected drop from 89.9% with all frequencies for 3 cycles (equivalent 48 timesteps) since we are only giving partial information for reconstruction.

As an additional experiment to re-emphasize that our ordering is beneficial, we give only the top 8 frequencies for the same number of cycles as the previous experiment (6 cycles, 48 timesteps) and get 87.2%, which is 3% better than the scheme with intermittent frequencies, validating the importance of ordering timesteps.

