# OpenReview forum: "DCT-SNN: Using DCT to Distribute Spatial Information over Time for Learning Low-Latency Spiking Neural Networks"
_ICLR.cc/2021/Conference — Reject_

### Official Review · AnonReviewer4 · 2020-10-21
**his work utilizes DCT based encoding method into spiking neural networks. This encoding method combines the mathematics of DCT and bio-plausibility of spiking neurons (LIF) to enrich the correlations in the spatio-temporal domain of spikes, which is quite inspiring.**

**Rating:** 6
**Confidence:** 3

**Review:**

Authors applied this algorithm into several datasets such as CIFAR 10, CIFAR 100 and TinyImagenet, they argued the image classification accuracies onto these datasets were comparable. The experiments were sufficient but not well designed. The reported experimental results show some novelty and good performance.

In the early stage of the spiking neural networks, the encoding methods are very important, especially for the training process. As the authors said that the rated based encoding method brings much more time latency which is time consuming. Therefore, the topic of this work is critical. However I have some worries about the proposed methods, from my point of view, this work is just combing the DCT and ANN-SNN method, the novelty is significantly limited, but the idea is interesting. Then, the experimental results reported by this paper were not well designed. The authors argued that the rated based encoding methods are time consuming, but they just did not compare the much more temporal encoding methods in Table 3. And for me, CIFAR10, CIFAR 100 and MNIST are in the same quantity level, which means that you used a VGG net is waste of resource (VGG is too deep). I even did not know what parts of the final results works, the deep CNN based network architecture? The DCT encoding method? Or the ANN-SNN methods. Actually, ANN-SNN method is not a typical bio-inspired way to construct a SNN, I prefer to see you adopt the proposed method into a Tempotron or STDP based learning rule not a surrogate-gradient based rule. The computational efficiency is nice; especially the authors calculated the spike rate of each single layer, but if you just argued the proposed the method is energy consumption, you should at least consider the ANN training process, it is not a single trade-off between inference accuracy and latency. I have run the code from authors provided, the reproducibility is reliable.

Also there are some writing errors, such as thy->they, -s, etc. and reference missing, such as these important works:
1.	An FPGA Implementation of Deep Spiking Neural Networks for Low-Power and Fast Classification.
2.	Deep CovDenseSNN: A hierarchical event-driven dynamic framework with spiking neurons in noisy environment

---

> ### Author Response · Authors · 2020-11-19
> **Response to AnonReviewer4 (Part 1)**
>
> We thank the reviewer for their time and comments. We address some of the issues the reviewer mentioned here.  The revised version of the manuscript is uploaded with the changes highlighted in red.
>
> 1. However I have some worries about the proposed methods, from my point of view, this work is just combing the DCT and ANN-SNN method, the novelty is significantly limited, but the idea is interesting.
>
> Ans. Thank you for your comments. We wish to point out that naively using DCT and an ANN to SNN conversion method, we would get what we refer to as the SNN-d network in Table 2. It is a network trained on Poisson encoded spikes, initialized from an ANN trained on DCT coefficients. However, our method to distribute spikes over time involves an inverse transform, that takes us back to the pixel domain. We believe our novelty lies in that we distribute reconstruction information over time in a ranked, orthogonal manner. The orthogonality allows us to offer non-interfering refinement over time and the ordering allows us to trade-off accuracy for inference latency in a principled manner. Moreover, as can be seen from Table 2, the proposed DCT-SNN outperforms SNN-d (which is just the combination of DCT and ANN-SNN method).
>
> 2. Then, the experimental results reported by this paper were not well designed. The authors argued that the rated based encoding methods are time consuming, but they just did not compare the much more temporal encoding methods in Table 3.
>
> Ans. We thank the reviewer for pointing us towards more temporal methods. We already compared our work with time-to-first-spike (TTFS) (Park et al., 2020) and phase-coding (Kim et al., 2018) in Table 3. To further elucidate the benefits of DCT-SNN over these methods, we report the inference accuracy vs timesteps curves for these algorithms along with other rate-based schemes in Fig. 9 of appendix A.5.2, adopted from Fig. 6 of (Park et al., 2020).
>
> We would like to add that most other temporal encoding methods only show results for MNIST. For further comparison with these works, we simulate our method on MNIST on a shallow 1 layer network (784-100-10 neurons, all fully connected) and show the comparison with several reported temporal encoding mechanisms in Table 4, Appendix Section A.5.1. S4NN (Kheradpisheh et al. 2020) uses temporal backpropagation for spiking neural networks with one spike per neuron and achieves 97.4% accuracy on MNIST with 256 timesteps. Our proposed method obtains 98.54% with just 16 timesteps. Comsa et al. (2020) report 97.9% accuracy on MNIST by using temporal coding, however the required timesteps are not mentioned. Another recent work (Stephan et al. 2020) also performs supervised learning in a temporal manner to obtain 85% accuracy on MNIST using just 10 timesteps. Yu et al. (2013) employ a tempotron-based supervised temporal learning scheme and achieve 78% on MNIST with 100 timesteps. Another tempotron-based work (Xu et al. 2018) reports 87% accuracy with an augmented framework with perceptron-inception, and they do not report timesteps for convergence. Again, a temporal STDP-based learning was used in Beyeler et al. (2013) which was able to reach 91.6% accuracy on MNIST with 500 timesteps. Notably, DCT-SNN outperforms all these methods as shown in Table 4, Appendix Section A.5.1. We emphasize that our network can converge even with 2 timesteps, which is considerably lower compared to these other works.

---

> > ### Author Response · Authors · 2020-11-19
> > **Response to AnonReviewer4 (Part 2)**
> >
> > 3.  CIFAR10, CIFAR 100 and MNIST are in the same quantity level, which means that you used a VGG net is waste of resource (VGG is too deep). I even did not know what parts of the final results works, the deep CNN based network architecture? The DCT encoding method? Or the ANN-SNN methods.
> >
> > Ans. Regarding the choice of datasets: We agree with the reviewer that CIFAR10, CIFAR 100 and MNIST are on the same level in terms of image size. However, the complexity for drawing decisive boundaries is much higher in CIFAR than in MNIST, since MNIST has grayscale images of digits that are easily separable with just fully connected layers with high degree of accuracy. However, CIFAR consists of 3 channel colored images with more varied classes than just digits. This needs convolutional layers to function well. CIFAR100 consists of 100 classes and requires deeper architectures to work well.
> >
> > Role of deeper architectures:  We trained a shallower DCT-SNN with 2 convolution layers (20 channels with kernel 5 × 5 - 30 channels 5 × 5), 2 × 2 average-pooling layers after each convolution layer, followed by 2 fully connected layers (256 and 10 neurons, respectively) on CIFAR10 and this achieves 78.1% accuracy. As shown in Table 1, VGG5 achieves 83.5% and VGG9 reaches ~90%.  Wu et al. (2019) confirm that accuracy improves with deeper nets. For this reason, we believe that deeper nets are needed to achieve competitive accuracy. Therefore, similar to many of the works shown in Table 3 in our manuscript, we use VGG style architectures to test our accuracy.
> >
> > Role of training methods: The ANN-SNN conversion methods are required for faster convergence of training on large-scale datasets. As explained in Rathi et al. (2020), training SNNs from scratch for more complex datasets such as CIFAR-100, Tiny-Imagenet etc. is both time-consuming and power-intensive. With a pretrained ANN as an initialization, the SNN training is able to converge within 15-20 epochs as opposed to ~200 epochs required for SNNs trained from scratch. Again, training  ANNs is much faster than training one epoch of an iso-architecture SNN, since the SNN training involves backpropagation through numerous timesteps. Though ANN-SNN enables comparable SNN accuracy for large-scale datasets, this method still suffers from high inference latency (Rueckauer et al. 2017, Sengupta et al. 2019).
> >
> > We utilize the benefits of all these methods in our work, but focus in particular on reducing latency with good architectures on middle scale datasets with competitive accuracies. We show that a baseline that also uses all these methods still suffers from high latency, and demonstrate that keeping everything else apart from the encoding part constant, we are able to reduce the convergence timesteps by more than half (Fig. 5 in our manuscript). So to summarize, depth enables achieving maximum accuracy, ANN-SNN converted networks with post-conversion SNN training methods aid in faster training convergence compared to training SNNs from scratch, and our DCT based encoding scheme results in reduced inference latency.
> >
> > 4. ANN-SNN method is not a typical bio-inspired way to construct a SNN, I prefer to see you adopt the proposed method into a Tempotron or STDP based learning rule not a surrogate-gradient based rule.
> >
> > Ans. We thank the reviewers for the comment. We agree that STDP and Tempotron (Robert and Sompolinsky (2006)) are more biologically plausible methods for training SNNs. However, no work has currently been able to show competitive learning ability on even middle scale datasets such as CIFAR10 with deeper networks (Diehl and Cook (2015)). Most bio-plausible learning methods show results primarily for MNIST. The focus in our manuscript is to introduce an encoding scheme that targets the inference latency of SNNs. Ideally, the encoding scheme is independent of the training methodology and we do not see any reason why the encoding scheme would fail with bio-plausible learning mechanisms that are shown to work with Poisson encoding. We appreciate the direction and will explore it in the future work.
> >
> > 5. The computational efficiency is nice; especially the authors calculated the spike rate of each single layer, but if you just argued the proposed the method is energy consumption, you should at least consider the ANN training process, it is not a single trade-off between inference accuracy and latency.
> >
> > Ans. We thank the reviewer for his comment, however we are primarily interested in inference latency for two reasons: training cost is a single time cost, and it can be performed on an expensive cloud machine before deployment on a resource constrained device. For this reason, in our manuscript, we only focus on inference latency and do not take into account training costs for any method.

---

> > > ### Author Response · Authors · 2020-11-19
> > > **Response to AnonReviewer4 (Part 3)**
> > >
> > > 6.. Reference missing, such as these important works:
> > > An FPGA Implementation of Deep Spiking Neural Networks for Low-Power and Fast Classification.
> > > Deep CovDenseSNN: A hierarchical event-driven dynamic framework with spiking neurons in noisy environment
> > >
> > > Ans. We thank the reviewer for pointing us to these works, and have added them to the revised manuscript in the relevant sections.
> > >
> > >
> > > -------------------------------------------------------------------------------------------------------------------------------------------------------------------------------
> > >
> > > References:
> > >
> > > Diehl, Peter U., and Matthew Cook. "Unsupervised learning of digit recognition using spike-timing-dependent plasticity." Frontiers in computational neuroscience 9 (2015): 99
> > >
> > > Wu, Yujie, et al. "Direct training for spiking neural networks: Faster, larger, better." Proceedings of the AAAI Conference on Artificial Intelligence. Vol. 33. 2019.
> > >
> > > Rathi, Nitin, et al. "Enabling deep spiking neural networks with hybrid conversion and spike timing dependent backpropagation." arXiv preprint arXiv:2005.01807 (2020).
> > >
> > > Gütig, Robert, and Haim Sompolinsky. "The tempotron: a neuron that learns spike timing–based decisions." Nature neuroscience 9.3 (2006): 420-428.
> > >
> > > Bodo Rueckauer, Iulia-Alexandra Lungu, Yuhuang Hu, Michael Pfeiffer, and Shih-Chii Liu.Conversion of continuous-valued deep networks to efficient event-driven networks for imageclassification.Frontiers in neuroscience, 11:682, 2017.
> > >
> > > Abhronil Sengupta, Yuting Ye, Robert Wang, Chiao Liu, and Kaushik Roy. Going deeper in spikingneural networks: Vgg and residual architectures.Frontiers in neuroscience, 13:95, 2019.
> > >
> > > Park, Seongsik, et al. "T2FSNN: Deep Spiking Neural Networks with Time-to-first-spike Coding." arXiv preprint arXiv:2003.11741 (2020).
> > >
> > > J. Kim et al., “Deep neural networks with weighted spikes,” Neurocomputing, vol. 311, pp. 373–386, 2018
> > >
> > >  Kheradpisheh, S. R., & Masquelier, T. (2020). S4NN: temporal backpropagation for spiking neural networks with one spike per neuron. International Journal of Neural Systems, 30(6), 2050027.
> > >
> > > Comsa, I. M., Fischbacher, T., Potempa, K., Gesmundo, A., Versari, L., & Alakuijala, J. (2020, May). Temporal coding in spiking neural networks with alpha synaptic function. In ICASSP 2020-2020 IEEE International Conference on Acoustics, Speech and Signal Processing (ICASSP) (pp. 8529-8533). IEEE.
> > >
> > > Stephan, A., Gardner, B., Koester, S. J., & Gruning, A. (2020). Supervised Learning in Temporally-Coded Spiking Neural Networks with Approximate Backpropagation. arXiv preprint arXiv:2007.13296.
> > >
> > > Qiang Yu, Huajin Tang, Kay Chen Tan, and Haizhou Li. Rapid feedforward computation by temporal encoding and learning with spiking neurons.IEEE transactions on neural networks and learning systems, 24(10):1539–1552, 2013.
> > >
> > > Qi Xu, Yu Qi, Hang Yu, Jiangrong Shen, Huajin Tang, and Gang Pan. Csnn: An augmented spiking based framework with perceptron-inception. InIJCAI, pp. 1646–1652, 2018.
> > >
> > > Michael Beyeler, Nikil D Dutt, and Jeffrey L Krichmar. Categorization and decision-making in a neurobiologically plausible spiking network using a stdp-like learning rule. Neural Networks, 48:109–124, 2013.

---

### Official Review · AnonReviewer2 · 2020-10-26
**Novel input spike encoding method for SNN but lacks crucial experiments to support the conclusion**

**Rating:** 6
**Confidence:** 4

**Review:**

This paper proposes an encoding method based on the Discrete Cosine Transform (DCT) for Spiking Neural Network (SNN). The key idea is to decompose an image into different frequency components and feed them to the SNN sequentially. Compared to the Poisson coding method used in most SNN studies, the proposed encoding method significantly decreases the latency that the SNN needs for image classification while having minimal accuracy decease.

Highlights:

1. The idea of using DCT for input spike encoding is novel and has great potential. One of the problems that prevent the SNN from using fewer inference timesteps is the ineffectiveness of encoding input information. Using DCT, the method can potentially filter out less important information and more effectively encode the information in limited timesteps (as shown in Fig. 6 and Fig. 8 in the paper).

2. The paper doesn't directly learn in the frequency domain generated from DCT. Instead, it reverse transforms the DCT result back to the spatial domain and spreads it into different timesteps of the SNN. By doing so, the spike encoding gives more importance to the low-frequency information in the image. This is desirable because low-frequency information is more important than high-frequency information in the image for classification.

Concerns:

1. The paper lacks experiments to show that DCT directly contributes to the decrease of timesteps for classification. Although comparisons with earlier SNN works that use Poisson encoding are shown, there is a lack of comparison with any SNN methods that directly convert pixel values into spikes using IF neurons and threshold selection. Thus, the existing experiments are not sufficient to exclude the possibility that the latency decrease is not due to DCT. The reviewer suggests conducting additional experiments for this.

2. While the proposed method only focuses on the input encoding of SNN, many recent papers target new training methods (such as [Jibin Wu et al, 2019], [Sen Lu et al, 2020]) that also result in significant latency decrease of SNN for image classification. The paper lacks experiments to compare the performance with these more recent results. The reviewer suggests conducting additional experiments for this.

3. The paper claims that the proposed method has better performance than ANNs trained on DCT coefficients. However, this is not a fair comparison since their encodings are different. The ANNs trained on DCT coefficients (such as [Max Ehrlich et al, 2019]) follow the same procedure as JPEG compression. The encoding uses non-overlapping 8x8 blocks, and the ANNs directly learn from the JPEG transformed domain. If the paper wants to compare with these ANNs, it needs experiments using the same encoding input.

4. The example in Fig. 1 for the reverse transformation is not the same as the source code (spike_model_vgg9_submit.py: Line 233). In section 3.1, the paper performs inverse transform by doing an element-wise multiplication between the transformed vector Y and each frequency basis in the transformation matrix, and claims the same method generalizes to the 2D case. However, the source code performs inverse transformation using the particular element in the transformed matrix Y (it's now a matrix but not a vector) corresponding to the specific frequency basis. Thus, the explanation in the paper contradicts the implementation. The reviewer thinks the example given in Fig. 1 is mathematically incorrect.

5. The proposed method spreads the reverse transformed image into different timesteps, and each timestep corresponds to a particular frequency. However, the paper doesn't explore other possible approaches for spreading the information. For example, the encoding method can use multiple subsequent timesteps for a particular frequency or only present intermittent frequencies. The lack of in-depth analysis of the proposed method possibly prevents the paper from fully exploring the potential of the use of DCT encoding for SNN.

Minor Comments:

1. What is the meaning of "ov" in Fig. 4? The reviewer thinks it means "overlap". However, it needs to be explained in the text or figure caption.

2. In Table 2, it's not clear whether DNN-d uses DCT coefficients or the reverse transformed image. If the DNN-d uses the DCT coefficient, is there any change to the ConvNet since the DCT destroys the block's spatial relationships?

3. In Table 2, the SNN-d results for TinyImageNet are missing. Is there any reason for that?


Jibin Wu et al, 2019, A Tandem Learning Rule for Effective Training and Rapid Inference of Deep Spiking Neural Networks

Sen Lu et al, 2020, Exploring the Connection Between Binary and Spiking Neural Networks

Max Ehrlich et al, 2019, Deep residual learning in the jpeg transform domain

 Since most of my primary concerns are resolved, I have updated my rating based on the revised version.

---

> ### Author Response · Authors · 2020-11-19
> **Response to AnonReviewer2 (Part 1)**
>
> We wish to thank the reviewer for the in-depth review and pointing us to relevant literature. It has helped us further improve our paper and we address individual comments here.  The revised version of the manuscript is uploaded with the changes highlighted in red.
>
> 1. The paper lacks experiments to show that DCT directly contributes to the decrease of timesteps for classification. Although comparisons with earlier SNN works that use Poisson encoding are shown, there is a lack of comparison with any SNN methods that directly convert pixel values into spikes using IF neurons and threshold selection. Thus, the existing experiments are not sufficient to exclude the possibility that the latency decrease is not due to DCT. The reviewer suggests conducting additional experiments for this.
>
> Ans. We thank the reviewer for the question. We have not come across many works in our literature survey that directly expose the input pixel intensities to an IF/LIF neuron. As an alternative, we discuss some works that expose analog values of the pixels directly to the first convolutional layer, instead of spikes.
>
> The first of these is a work that the reviewer directed us to: Jibin Wu et al, (2019). These authors appear to use an encoding scheme that converts input pixels directly to spikes, but we were unclear about some details (discussed as part of the rebuttal of the next comment). We were unable to simulate their results since the code is not available and there was not enough time to reproduce the implementation.
>
> Next, in Table 3 in our main text, we compare with 2 works,  Rueckauer et al. (2017) and Lu and Sengupta. (2020) that expose the analog pixel intensities directly as input currents to the first layer (instead of spike trains) and hence perform analog (ANN-like) computations at the first layer. Compared to Rueckauer et al. (2017), we converge to slightly better accuracy at nearly an order of magnitude lesser timesteps (400 vs 48) and compared to Lu and Sengupta. (2020), we use a smaller network, but still converge to ~5% higher accuracy at slightly fewer timesteps (62 vs 48).
>
> Wu et al. (2019) also expose the pixels directly to first layer and the spike-train is generated through IF/LIF neurons after this layer . They achieve 90.53% accuracy on CIFAR-10 using just 12 timesteps on a network with 5 convolutional and 2 fully connected layers. We tried to implement this algorithm ourselves, since the source code was not public. We upload our version of their code along with our supplementary material under the name wu_direct_training.ipynb. However, we could not reproduce the results reported, since the hyperparameters of implementation are not provided in the paper, such as learning rate, decay constant and ‘v ‘parameter in Eqn. (9) of the paper. Additionally, in Wu et al. (2019), after each conv layer, the binary activations go through a channel-wise normalization (termed as neunorm in the paper). This makes the 0/1 activations essentially analog, as can be seen from Eqn. (9) and (10) in the paper.  We are not clear whether the efficiency of 12 timesteps is arising due to their encoding scheme, because of the proposed channel-wise normalization resulting in analog computation at each layer, or their voting scheme based on class-wise populations at the output. We believe that the analog nature of computation makes this network closer to ANNs than SNNs, resulting in the significant reduction in timesteps, especially since both Jibin Wu et al. (2019) and Wu et al. (2019) claim that they can converge to a good accuracy in a single timestep.
>
> We sincerely thank the reviewer for helping us show our work in better context and have appended these results to Table 3, and have included this discussion in the main text in Section 4.

---

> > ### Author Response · Authors · 2020-11-19
> > **Response to AnonReviewer2 (Part 2)**
> >
> > 2. While the proposed method only focuses on the input encoding of SNN, many recent papers target new training methods (such as [Jibin Wu et al, 2019], [Sen Lu et al, 2020]) that also result in significant latency decrease of SNN for image classification. The paper lacks experiments to compare the performance with these more recent results. The reviewer suggests conducting additional experiments for this.
> >
> >
> > Ans. The authors wish to thank the reviewer for directing us to Jibin Wu et al, (2019). It is indeed a novel and promising way to train SNNs in conjunction with ANNs. However, we believe that the primary focus of that work is a training mechanism, and our work focuses entirely on the encoding scheme, which is orthogonal to the training mechanism. We can apply Jibin Wu et al.’s training mechanism to our encoding scheme as well. Additionally, they make many different modifications in their method, including introduction of Batch-Norm layers, a non-Poisson encoding scheme, and a training methodology with ANN and SNN sharing weights. We are not entirely clear which modification contributes in what proportion to the achievement of low inference timesteps. We are also unclear about their neural encoding layer, since it appears to be exposing pixels directly to the IF neuron, but still requires a gradient according to Figure 4 in their paper. Since their code has not been made publicly available, we were unable to simulate their results or clarify our confusion. We further note that their accuracy is about 1% lower than ours on CIFAR-10 using LIF neurons (89.04% for theirs vs 89.94% for ours). Another point of distinction is that we can train DCT-SNN both from scratch and locally using bio-plausible techniques like STDP, which Jibin Wu et al. cannot do since their SNN is trained conjoined to an ANN. However, we do believe that this is an interesting method and have added this reference, along with some more recent encoding mechanisms to Table 3 in the main text.
> >
> > In addition, we wish to bring to notice that we already incorporate suggestions made by Lu and Sengupta, (2020) in our network, in the form of using a percentile of layerwise activation distribution as that layer’s threshold as opposed to the maximum value. We also used avgpool instead of maxpool since Lu and Sengupta show that it leads to lower accuracy degradation during ANN-SNN conversion. However, ours is primarily an encoding scheme, and using that on top of their architectural modifications gives us an improvement in accuracy and latency, as can be seen in Table 3, in the form of 4% higher accuracy in slightly fewer timesteps for CIFAR100.
> >
> > 3. The paper claims that the proposed method has better performance than ANNs trained on DCT coefficients. However, this is not a fair comparison since their encodings are different. The ANNs trained on DCT coefficients (such as [Max Ehrlich et al, 2019]) follow the same procedure as JPEG compression. The encoding uses non-overlapping 8x8 blocks, and the ANNs directly learn from the JPEG transformed domain. If the paper wants to compare with these ANNs, it needs experiments using the same encoding input.
> >
> > Ans. We have shown our results with 8x8 non overlapping DCT blocks in Figure 4 in the main text, and show that we get 80.7% accuracy. But in our transform, we do go back to the pixel domain with the reverse transform.  So, for a fairer comparison to ANNs trained in the frequency domain with JPEG style blocks, we simulated an SNN version of VGG5 with 8x8 DCT, without overlap, trained directly on DCT coefficients using the Poisson encoding scheme referred to as SNN-d in the text. We first note that the base ANN we train for conversion outperforms the ANNs mentioned in the literature, getting 84% accuracy on CIFAR10 (in contrast,  Ehrlich et al, (2019) achieve 72.5%). When we train the SNN on Poisson encoded DCT coefficients, we achieve 81.9% accuracy.  We thank the reviewer for pointing the differences out and have edited the corresponding discussion in the text to clarify the comparison.

---

> > > ### Author Response · Authors · 2020-11-19
> > > **Response to AnonReviewer2 (Part 3)**
> > >
> > > 4. The example in Fig. 1 for the reverse transformation is not the same as the source code (spike_model_vgg9_submit.py: Line 233). In section 3.1, the paper performs inverse transform by doing an element-wise multiplication between the transformed vector Y and each frequency basis in the transformation matrix, and claims the same method generalizes to the 2D case. However, the source code performs inverse transformation using the particular element in the transformed matrix Y (it's now a matrix but not a vector) corresponding to the specific frequency basis. Thus, the explanation in the paper contradicts the implementation. The reviewer thinks the example given in Fig. 1 is mathematically incorrect.
> > >
> > > Ans. We sincerely thank the reviewer for pointing out the error in the figure and apologize for this oversight. We have fixed both Figs. 1 and 2 and the associated explanation to point out that it is a matrix multiplication instead of a Hadamard product. We wish to emphasize that our results are still correct, and hold. The implementation was based on reconstruction with matrix multiplication (corresponding to the updated figure), and hence, our results remain unaltered.
> > >
> > > 5. The proposed method spreads the reverse transformed image into different timesteps, and each timestep corresponds to a particular frequency. However, the paper doesn't explore other possible approaches for spreading the information. For example, the encoding method can use multiple subsequent timesteps for a particular frequency or only present intermittent frequencies. The lack of in-depth analysis of the proposed method possibly prevents the paper from fully exploring the potential of the use of DCT encoding for SNN.
> > >
> > > Ans. We thank the reviewer for the question, and we did experiment with the way we reconstruct the image over time. However, we decided not to include those experiments because of the page limit, and because the scheme described in the paper performed the best. We list the experiments here, and also add them to Appendix section A.6
> > >
> > > (a) Since we knew 3 cycles (48 timesteps) gave the best results for VGG9/CIFAR10, instead of giving frequencies cyclically for 3 cycles, we tried interleaving the frequencies; i.e. instead of frequencies being input in the order 0,1,2,…15,0,1,2,….15,0,1,2,…15, we input them as 0,0,0,1,1,1,2,2,2, ….15,15,15. The original scheme gave the best reported accuracy of 89.94% and the interleaved scheme only achieved 79.7%.
> > >
> > > (b) A similar cyclic vs interleaving experiment was done with the frequencies limited to top 8, repeated for 3 cycles (24 timesteps). The cyclic scheme achieved 69.7% and the interleaved achieved 53.73%. We believe this drop is due to the resetting of membrane potential as it fires between timesteps, causing temporal dependency to be incorporated between timesteps. Interleaving cannot leverage this dependency since the same information is repeated for multiple consecutive timesteps. Additionally, since the earlier DCT coefficients contain most of the energy, the spikes at the later timesteps start dying out with interleaved frequencies.
> > >
> > > (c) We also experimented with giving intermittent frequencies such as (0,2,5,7,9,10,12,15) given cyclically for 6 cycles (48 timesteps) and got 84.42%, an expected drop from 89.94% with all frequencies for 3 cycles (equivalent 48 timesteps) since we are only giving partial information for reconstruction.
> > >
> > > (d) As an additional experiment to re-emphasize that our ordering is beneficial, we give only the top 8 frequencies for the same number of cycles as the previous experiment (6 cycles, 48 timesteps) and get 87.2%, which is ~3% better than the scheme with intermittent frequencies from c), validating the importance of ordering timesteps.
> > >
> > > Minor Comments:
> > >
> > > What is the meaning of "ov" in Fig. 4? The reviewer thinks it means "overlap". However, it needs to be explained in the text or figure caption.
> > >
> > > We revised the manuscript mentioning that 'ov' refers to overlap
> > >
> > > In Table 2, it's not clear whether DNN-d uses DCT coefficients or the reverse transformed image. If the DNN-d uses the DCT coefficient, is there any change to the ConvNet since the DCT destroys the block's spatial relationships ?
> > >
> > > We did not make any changes to the architecture for working with DCT coefficients. Not making changes did not hurt since our parent ANN accuracy trained on DCT coefficients (ANN-d) shows competitive performance to ANN trained on pixels (ANN-p) in Table 2.

---

> > > > ### Author Response · Authors · 2020-11-19
> > > > **Response to AnonReviewer2 (Part 4)**
> > > >
> > > > In Table 2, the SNN-d results for TinyImageNet are missing. Is there any reason for that?
> > > >
> > > > Thank you for noticing that. From Table 2, we note that in case of TinyImageNet, the ANN trained on dct coefficients (ANN-d) that had the configuration for conversion (removing batchnorm and maxpool) performs considerably worse (~11%) than ANN trained on pixels (ANN-p). This means that our SNN trained on DCT coefficients (SNN-d) would similarly perform much worse than SNN-p. Moreover, usually the maximum accuracy the SNN can reach up to is that of the corresponding ANN and our DCT-SNN was already outperforming ANN-d in this case. As a result, we did not feel this result would add to the discussion, and simulating SNNs on large datasets requires considerable GPU time and resources, and hence we skipped this result. However, for completeness, we ran the experiment after the deadline and have added the number to the table, the SNN-d on Tiny-Imagenet achieves 44.6% with 250 timesteps.
> > > >
> > > >
> > > > --------------------------------------------------------------------------------------------------------------------------------------------------------------------
> > > >
> > > > References
> > > >
> > > >
> > > > Jibin Wu et al, 2019, A Tandem Learning Rule for Effective Training and Rapid Inference of Deep Spiking Neural Networks
> > > >
> > > > Rueckauer, B., Lungu, I. A., Hu, Y., Pfeiffer, M., & Liu, S. C. (2017). Conversion of continuous-valued deep networks to efficient event-driven networks for image classification. Frontiers in neuroscience, 11, 682.
> > > >
> > > > Lu, S., & Sengupta, A. (2020). Exploring the Connection Between Binary and Spiking Neural Networks. arXiv preprint arXiv:2002.10064.
> > > >
> > > > Wu, Yujie, et al. "Spatio-temporal backpropagation for training high-performance spiking neural networks." Frontiers in neuroscience 12 (2018): 331
> > > >
> > > > Max Ehrlich et al, 2019, Deep residual learning in the jpeg transform domain

---

> > ### Comment · AnonReviewer2 · 2020-11-23
> > **Comparison with the direct input encoding method**
> >
> > Thanks for addressing my concerns about the paper in the revised version of the manuscript. I'm delighted that my review can help the authors to improve their paper. Since most of my primary concerns are resolved, I have updated my rating based on the revised version.
> >
> > One more question regarding Part 1 of the authors' response: There are many recent papers on SNN using the direct input encoding method that learns the first layer of the SNN as a spike encoder (for example, an ICLR paper submitted this year: https://openreview.net/forum?id=u_bGm5lrm72). Can the authors discuss the proposed encoding method's differences and advantages against the direct input encoding?

---

> > > ### Author Response · Authors · 2020-11-24
> > > **Response to AnonReviewer2: Comparison with direct input encoded hybrid SNNs**
> > >
> > > We are grateful to the reviewer for updating the rating, and thank them for pointing us to an additional reference. Conceptually, the works mentioned use the first layer of the SNN as the spike encoder taking the analog pixels as input, and expose the activation map generated after the first convolution to the IF/LIF neurons. This makes the computations at the first layer of the SNN same as the corresponding ANN. In contrast, our method generates spikes at the input, which represent pixel values before passing them to the SNN and the first layer performs the convolution on spikes. In our opinion, the direct encoding methods can be visualized as a hybrid of ANN and SNN representations for learning, since the 1st layer of the SNN no longer operates on the spike representation of pixels. We believe that such a scheme can be taken further into the second layer for example, by letting the first two layers be calculated like the ANN and then exposing the activation map generated from the second convolutional layer to the spike generator. In our work, the input to the first layer of the SNN is a spike representation in the pixel domain itself, and the first layer feature extraction through convolution is performed on the spikes, unlike ANN or the direct input encoded hybrid SNNs. If we visualize a spectrum of computation being spread upon timesteps, SNNs and ANNs lie on either end. Standard SNNs, including ours, distribute the pixel information across time and utilize timesteps to perform all computations, whereas ANNs use a single timestep and perform all computation in one shot. The hybrid scheme calculates some layers in an ANN fashion, extracting features from analog inputs, and computes the remaining layers in SNN fashion by spreading the pre-extracted features over time steps, and therefore would lie in the middle of the spectrum. Additionally, these hybrid networks keep the same pixels as input for all timesteps. In contrast, we offer non-interfering information per timestep in a ranked manner.

---

### Official Review · AnonReviewer1 · 2020-10-29
**This paper discusses a method to make spiking networks more relevant to latency-sensitive applications on the edge. I believe the authors' method is relevant to this problem, but doesn't feel like it uncovers anything fundamentally new.  It is an engineering solution to a specific problem; and spiking networks are not a widely used method at the edge currently.**

**Rating:** 6
**Confidence:** 5

**Review:**

The scheme proposed breaks down the information in a block of an image into orthogonal basis functions (DCT is used) to make a progressively better reconstruction of the original image block with the addition of more basis functions used (like an nth order Taylor expansion).  The increasing spatial frequency components are known to be perceptually less sensitive (they need to include this) in images, so the low freq components can be presented first.  Each freq component is encoded into spikes sequentially, thereby staging the more perceptually important information first, with less important info coming later.  This reorders the presentation of information to allow a tradeoff of image quality with time/latency.

I think the solution proposed is well-founded and will indeed mitigate the latency problem for spiking neural networks.  However, this feels a bit more like an engineering solution to a specific problem rather than a new concept.  I do like the injection of methods from other fields like image/video compression; it often feels that the deep learning field rediscovers things that have been uncovered years ago in other fields.  I see that as the main value of the paper in addition to helping to make spiking neural networks a POSSIBLE viable solution to edge deployment.

Section 1: I don’t think it’s a strongly supported claim that deep learning architectures are unsuitable for edge deployment.  There are plenty in deployment and there are new processors (Movidius, Mythic, etc) that can handle these computations for real-time applications.  I’d suggest a softer language there.  This does weaken the motivation for the paper though.

Section 1, second paragraph:  typo: Thy -> The

Section 3.2: On constraints for the transforms.  Did the authors consider Integer Transform (IT)?  This is used in MPEG/AVC.  It is a reversible transform that is an integer simplification of the DCT.  Given that the point of the paper is to decrease latency and computing requirements for edge deployments, this could help.

Section 3.2: The authors do a good job of sweeping performance for different block sizes.

Figure 5:  Isn’t it an obvious result that more time steps are required for Poisson vs DCT?  There simply aren’t enough bins to sum over to have a result until a certain point.

---

> ### Author Response · Authors · 2020-11-19
> **Response to AnonReviewer1**
>
> We thank the reviewer for their time and for pointing us to a transform that can further enhance our efficiency. We have incorporated the suggested changes in the text and address individual comments here.  The revised version of the manuscript is uploaded with the changes highlighted in red.
>
> 1. Section 1: I don’t think it’s a strongly supported claim that deep learning architectures are unsuitable for edge deployment. There are plenty in deployment and there are new processors (Movidius, Mythic, etc) that can handle these computations for real-time applications. I’d suggest a softer language there. This does weaken the motivation for the paper though.
>
> Ans. We thank the reviewer for the comments. We have softened our language to accommodate custom solutions in Section 1. However, we believe that the bio-plausibility and the sparse event driven nature of SNNs leading to improved energy efficiency still makes them attractive for edge deployment.
>
> 2. Section 1, second paragraph: typo: Thy -> The
>
> Ans. Thanks for pointing this, we have fixed it.
>
> 3. Section 3.2: On constraints for the transforms. Did the authors consider Integer Transform (IT)? This is used in MPEG/AVC. It is a reversible transform that is an integer simplification of the DCT. Given that the point of the paper is to decrease latency and computing requirements for edge deployments, this could help.
>
> Ans.  We thank the reviewer for pointing us to Integer Transforms (IT) used in MPEG and AVC encoding. IT is an approximation of DCT, and meets the constraints of orthogonal bases and ordering of timesteps we enforce in our method. Since we perform an inverse transform as well, and the IT is invertible, we do not expect to see any accuracy degradation with it. We simulated VGG9 with CIFAR-10 with IT instead of DCT as the choice of transform and found comparable convergence (89.2% for IT compared to 89.94% for DCT) for the same number of timesteps (48), corroborating our assumption. The ease of implementing IT in the form of just shift and add operations would add to the efficiency of our algorithm even more, bringing the factor of 4.6 for encoder operations further down (see Eqn. (4) in our manuscript). We have added the results for IT in the computational efficiency part of Section 4 of the revised manuscript.
>
> 4. Figure 5: Isn’t it an obvious result that more time steps are required for Poisson vs DCT? There simply aren’t enough bins to sum over to have a result until a certain point.
>
> Ans. We agree with the reviewer, but we hoped to emphasize using figure 5 that we report fully saturated results very early on. In comparison, the Poisson method keeps increasing its accuracy with more time steps. We also highlight through the figure that for the timesteps where we are almost converged, the Poisson method gives random accuracy. We hope that it makes a compelling case as to why our encoding scheme is a highly preferable alternative to the widely used Poisson encoding scheme.

---

> > ### Comment · AnonReviewer1 · 2020-11-21
> > **Clarity on motivation**
> >
> > Thanks for incorporating the updates.  My final comments are around making it clear why this work is relevant.  My rating of the work is a function of relevance...why try to solve a problem with spiking networks?  Is this really a problem that any practionners have? Being very compelling on this would certainly make for a more meaningful result.

---

> > > ### Author Response · Authors · 2020-11-21
> > > **Response to AnonReviewer1 on the Practical Utility of SNNs (Part 1/2)**
> > >
> > > Response:
> > >
> > > Thank you for your response. Your concern is valid, and we try to address it from two directions, namely the advantages that SNNs have over ANNs, and the realization of SNNs in practical use cases currently. We add a few lines to the same in the text, highlighted in blue.
> > >
> > > To preface our answer, we would like to emphasize that SNNs are not just a theoretical construct. The primary advantage (among others) is energy-efficiency that comes from sparse event driven nature of computation and the need for simple compute – accumulation (ACC) rather than multiply and accumulate (MAC) operation (known to be power hungry), required for ANNs.  IBM designed a non-commercial processor ‘TrueNorth’ (Akopyan et al.), and Intel designed it’s equivalent ‘Loihi’, (Davies et al.) that can train and infer on SNNs. The primary bottleneck with SNNs is that they are not efficient when run on GPUs, but when run on custom hardware, they have been benchmarked to be 109x more efficient in inference energy than GPUs (Blouw et al.) for the task of keyword spotting, while maintaining the same accuracy. The authors also show that Loihi is 5.3x more efficient than Movidius, and 23x more efficient than a CPU.
> > >
> > >
> > >
> > > Advantages offered by SNNs:
> > >
> > > 1. Bioplausibility: The brain is known to operate on spikes, resulting in event driven processing in the brain’s neural circuitry. While we do not want to blindly emulate the brain, we view it as existential proof that spike driven processing is a potential means to being able to perform tasks without consuming MegaWatts of power as consumed in GPUs.
> > >
> > > 2. Efficiency due to event driven nature: Since each spike is binary, computation is only performed when the neuron receives a positive spike. The spikes are sparse, implying that most of the network is inactive for most of the duration of inference, allowing for much lower power and energy consumption
> > >
> > > 3. Efficiency due to cheaper compute: Another benefit of spikes being binary, is that the weighted summation process $\Sigma_i w_i x_i$, which earlier involved a multiply and accumulate (MAC) operation, now becomes just an accumulate operation. The hardware is simplified, and the resulting accumulate operation is 5.1 times more efficient than the MAC on 45nm CMOS technology (Horowitz).
> > >
> > > A detailed discussion about the opportunities and challenges with SNNs is available in (Pfeiffer et al.)
> > >
> > >
> > > Practical Use Cases:
> > >
> > > 1. Work on designing hardware that can leverage event driven computation has started in earnest.  Non-commercial processors such as IBM's TrueNorth (Akopyan et al.) and Intel’s Loihi (Davies et al) are processors built to train and infer on SNNs and show state of the art accuracy while being more efficient than ANNs running on GPUs. (Cheng et al., Blouw et al.).
> > >
> > > 2. There have been spiking versions of well-known datasets like MNIST and CIFAR made by using neuromorphic event driven sensors (Serrano-Gotarredona et al., Li et al.). Recent works (Lee et al.) show that SNNs outperform ANNs in motion detection and navigation when the data is acquired via DVS (event driven) cameras.
> > >
> > > 3. SNNs have been shown to work well on more than just the vision and classification tasks, such as auditory modality (Pan et al.), gesture learning and online learning (Stewart et al), reinforcement learning (Patel et al.), time series data (Fang et al.), and navigation (Lee et al.)
> > >
> > >
> > > However, the requirement of large timesteps for inference reduces the advantages of power efficiency offered by event driven computation. This is an active area of research, and the problem we address to a significant degree in our manuscript.
> > >
> > > Personally, the authors believe that SNNs, though in their nascent stage, can be visualized as adding a time axis to binary ANNs (connection explored in Lu et al). From an input representation perspective, we can look at any analog value x, let’s say pixel intensity in the range of 0-255 as a summation of x spikes. The advantage of viewing it in this fashion, besides just efficient computation due to the binary nature of spikes and the event driven information processing, is that we can learn information from the timing between spikes as well. Data occurring in nature, as processed by humans, is always in the form of time series, and SNNs is an efficient way of utilizing that time dimension. Due to these reasons, we believe that exploring the potential of SNNs is a rewarding exercise.

---

> > > > ### Author Response · Authors · 2020-11-21
> > > > **Response to AnonReviewer1 on the Practical Utility of SNNs (Part 2/2)**
> > > >
> > > > References
> > > >
> > > > Serrano-Gotarredona, Teresa, and Bernabé Linares-Barranco. "Poker-DVS and MNIST-DVS. Their history, how they were made, and other details." Frontiers in neuroscience 9 (2015): 481.
> > > >
> > > > Lee, Chankyu, et al. "Spike-FlowNet: Event-based Optical Flow Estimation with Energy-Efficient Hybrid Neural Networks." arXiv preprint arXiv:2003.06696 (2020).
> > > >
> > > > Cheng, Hsin-Pai, et al. "Understanding the design of IBM neurosynaptic system and its tradeoffs: a user perspective." Design, Automation & Test in Europe Conference & Exhibition (DATE), 2017. IEEE, 2017.
> > > >
> > > > Davies, Mike, et al. "Loihi: A neuromorphic manycore processor with on-chip learning." IEEE Micro 38.1 (2018): 82-99.
> > > >
> > > > Pan, Zihan, et al. "An efficient and perceptually motivated auditory neural encoding and decoding algorithm for spiking neural networks." Frontiers in Neuroscience 13 (2019).
> > > >
> > > > Stewart, Kenneth, et al. "Online few-shot gesture learning on a neuromorphic processor." IEEE Journal on Emerging and Selected Topics in Circuits and Systems (2020).
> > > >
> > > > Fang, Haowen, et al "Multivariate time series classification using spiking neural networks." 2020 International Joint Conference on Neural Networks (IJCNN). IEEE, 2020.
> > > >
> > > > Pfeiffer, Michael, and Thomas Pfeil. "Deep learning with spiking neurons: opportunities and challenges." Frontiers in neuroscience 12 (2018): 774.
> > > >
> > > > Patel, Devdhar, et al. "Improved robustness of reinforcement learning policies upon conversion to spiking neuronal network platforms applied to Atari Breakout game." Neural Networks 120 (2019): 108-115.
> > > >
> > > > Lu, Sen, and Abhronil Sengupta. "Exploring the Connection Between Binary and Spiking Neural Networks." arXiv preprint arXiv:2002.10064 (2020).
> > > >
> > > > Li, Hongmin, et al. "Cifar10-dvs: an event-stream dataset for object classification." Frontiers in neuroscience 11 (2017): 309.
> > > >
> > > > Blouw, Peter, et al. "Benchmarking keyword spotting efficiency on neuromorphic hardware." Proceedings of the 7th Annual Neuro-inspired Computational Elements Workshop. 2019.
> > > >
> > > > F. Akopyan et al., "TrueNorth: Design and Tool Flow of a 65 mW 1 Million Neuron Programmable Neurosynaptic Chip," in IEEE Transactions on Computer-Aided Design of Integrated Circuits and Systems, vol. 34, no. 10, pp. 1537-1557, Oct. 2015, doi: 10.1109/TCAD.2015.2474396.
> > > >
> > > > Mark Horowitz. 1.1 computing’s energy problem (and what we can do about it). In2014 IEEEInternational Solid-State Circuits Conference Digest of Technical Papers (ISSCC), pp. 10–14.IEEE, 2014.

---

> > > > > ### Comment · AnonReviewer1 · 2020-11-23
> > > > > **relevance continued**
> > > > >
> > > > > Thanks for the citations.  I'm hoping my comments will nudge the discussion towards your last statement above ("Personally, the authors believe that SNNs, though in their nascent stage,...").  Being clear to the reader that SNNs hold promise, but nascent at the moment is important.
> > > > >
> > > > > Thank you.  My score will remain unchanged.

---

### Official Review · AnonReviewer3 · 2020-10-30
**A new coding scheme based on DCT for SNNs, but not convincing enough.**

**Rating:** 5
**Confidence:** 4

**Review:**

Pros:
1.	A novel coding scheme is proposed based on Discrete Cosine Transform (DCT) for efficient information expression in place of conventional Poisson distribution method. The required time-steps are 2-14x reduced compared with other conversion-SNNs or hybrid trained SNNs.
2.	DCT is data-independent while performing at par with PCA.
Cons:
1.	The experimental results are not convincing enough.
2.	Correctness Problem. I am afraid that the descriptions about the reconstruction of input image is wrong. The reverse transform should be a matrix multiplication instead of Hadamard product of the coefficients and the basis. For example, in Fig 1., X=Y_1*T_1+Y_2*T_2+…+Y_5*T_5. It results in conceptual errors in Fig1 & Fig2.
3.	The specific equation of DCT should be with the discussion of the desirable properties and constraints in the section of encoding scheme to provide a clear picture of the method.
About the experimental results:
1.	Could the authors try to provide some explanations for why DCT is able to outperform Poisson method or directly exposing the original image to the input spike neuron, since DCT’s reverse transform is a reconstruction of the original image over time?
2.	Fewer time-steps with relatively lower accuracy is kind of confusing. I would like that the authors could further show the required time-steps for reaching strictly equal (or better) accuracy results with other SNN works (especially those directly trained). It is because that the trade-off between accuracy and the number of time-steps is natural in SNNs. For example, Rathi et al. (2020) could increase their VGG16’s performance from 91.13% to 92.02% by adding 100 time-steps on CIFAR-10.
Rathi, Nitin, et al. "Enabling deep spiking neural networks with hybrid conversion and spike timing dependent backpropagation." arXiv preprint arXiv:2005.01807 (2020).

3.	I notice that you cite Wu's article in related works, but there is a lack of comparison to it in later experiment part. In my opinion, the results of the paper and its sequel on time-steps for training SNNs from scratch are worth-noticing. I would like a more comprehensive and fair comparison of results.
Wu, Yujie, et al. "Spatio-temporal backpropagation for training high-performance spiking neural networks." Frontiers in neuroscience 12 (2018): 331.
Wu, Yujie, et al. "Direct training for spiking neural networks: Faster, larger, better." Proceedings of the AAAI Conference on Artificial Intelligence. Vol. 33. 2019.

Clarity: The paper is fairly well-written.
Originality: DCT is a widely used transformation technique in signal processing and data compression, but this paper creatively explores it as a coding scheme in SNNs.
Significance: The proposed coding scheme might provide a new resolution to the high inference latency bottleneck in SNNs.

---

> ### Author Response · Authors · 2020-11-19
> **Response to AnonReviewer3 (Part 1)**
>
> We thank the reviewer for the insights and suggestions. They have helped us to improve the paper and the responses to the individual comments are given below. The revised version of the manuscript is uploaded with the changes highlighted in red.
>
> 1.  I am afraid that the descriptions about the reconstruction of input image is wrong. The reverse transform should be a matrix multiplication instead of Hadamard product of the coefficients and the basis. For example, in Fig 1., X=Y_1T_1+Y_2T_2+…+Y_5*T_5. It results in conceptual errors in Fig1 & Fig2.
>
> Ans. We sincerely thank the reviewer for pointing out the error in the figure and apologize for this oversight. We have fixed both Figs. 1 and 2 and the associated explanation to point out that it is a matrix multiplication instead of a Hadamard product. We wish to emphasize that our results are still correct, and hold. The implementation was based on reconstruction with matrix multiplication (corresponding to the updated figure), and hence, our results remain unaltered.
>
> 2. The specific equation of DCT should be with the discussion of the desirable properties and constraints in the section of encoding scheme to provide a clear picture of the method.
>
> Ans. Thank you for your comment. We have brought in the DCT equation from the appendix to the main text, where we talk about the constraints. Please refer to equations 1-3 for the DCT details.
>
> 3. Could the authors try to provide some explanations for why DCT is able to outperform Poisson method or directly exposing the original image to the input spike neuron, since DCT’s reverse transform is a reconstruction of the original image over time?
>
> Ans. The Poisson method uses the pixels as inputs but requires a large number of timesteps to accurately express the true value due to the randomization process. This process suffers from sampling errors; hence a long encoding time window is required to compensate for such errors. This error is large with a few steps but decreases with increasing time steps, whereas our method requires only one cycle (16 timesteps) to accurately express the image without error. Inputting a DCT frequency base per time step allows us to offer non-interfering information over time. The time steps are ordered by significance, which allows us to drop the non-important bases, and we do not have this ability when we directly input pixels, since pixels implicitly do not have any inherent ranking.
>
>  For further corroboration, Table 3 shows a comparison with two works that expose analog values of pixels directly to the first layer, instead of spikes, Rueckauer et al. (2017) and Lu and Sengupta (2020). As shown in the table, we converge to slightly better accuracy than Rueckauer et al. (2017) at nearly an order of magnitude lesser timesteps (400 vs 48). Compared to Lu and Sengupta. (2020), we use a smaller network (VGG9 vs VGG15), but still converge to ~5% higher accuracy at fewer steps (62 vs 48).

---

> > ### Author Response · Authors · 2020-11-19
> > **Response to AnonReviewer3 (Part 2)**
> >
> > 4. Fewer time-steps with relatively lower accuracy is kind of confusing. I would like that the authors could further show the required time-steps for reaching strictly equal (or better) accuracy results with other SNN works (especially those directly trained). It is because that the trade-off between accuracy and the number of time-steps is natural in SNNs. For example, Rathi et al. (2020) could increase their VGG16’s performance from 91.13% to 92.02% by adding 100 time-steps on CIFAR-10. Rathi, Nitin, et al. "Enabling deep spiking neural networks with hybrid conversion and spike timing dependent backpropagation." arXiv preprint arXiv:2005.01807 (2020).
> >
> > Ans. We would like to emphasize Figure 5 in the main text here, for training VGG9 on CIFAR10. The figure shows that we are reporting our results at saturated accuracy and increasing timesteps in our case does not result in any noticeable improvement. For context, the same figure provides the results of Poisson encoded SNNs, implemented using the hybrid conversion and Spike Timing Dependent Backpropagation method used in Rathi et al., (2020). It can be seen that in Rathi et. al.'s method, accuracy keeps increasing slowly with large number of timesteps, as the sampling error reduces. Since we pass all the information in 1 cycle, we are able to saturate to full accuracy at much lower timesteps.
> >
> > In addition, we show some comparisons with other works in Fig. 9, appendix section A.5.2 of the revised manuscript. The accuracy vs timestep values for recent methods are adopted from Fig. 6 of Park et al., (2020). The figure demonstrates the results of “T2FSNN” encoding scheme, which is a temporal encoding scheme and other rate and temporal encoding schemes such as “Rate” (Diehl et al., (2015) and Rueckauer et al. (2017)), “Phase” (Kim et al. (2018)), and, “Burst” coding (Park et al. (2019)). The left graph in Fig. 9, Appendix A.5.2 is recreated for CIFAR-10, and shows ~200 timesteps for the fastest convergence among these encoding methods, In contrast, we achieve ~90% accuracy in just 48 timesteps, saturating far earlier than any of these methods. From Fig. 6 of  Park et al. (2020), we can tell that the best version of “T2FSNN” first reaches 90% roughly at 240 steps, “Burst” at 300, “Phase” at 425, and “Rate” at 1200 timesteps, showing that we reduce latency by orders of magnitude, resulting in convergence at much fewer timesteps. The network is slightly different here, ours is VGG9 and the network used in Park et al. (2020) is VGG16, but in our opinion, that affects final convergence accuracy more than it affects orders of magnitude of timesteps for inference.
> >
> > Similarly, we exceed 68% accuracy in 48 timesteps when training VGG11 on CIFAR100. The graph on the right in Fig. 9, Appendix A.5.2 shows the convergence statistics for VGG16 on CIFAR100 using “T2FSNN”, “Burst”, “Phase” and “Rate”. The best version of “T2FSNN” reaches 68% roughly at 500 steps, “Burst” at 1500, “Phase” at 2000, and “Rate” does not go above 60% in even 3000 timesteps.
> >
> > We believe that these results show the efficacy of our encoding scheme in fast and accurate convergence compared to many encoding schemes published recently. These works were already mentioned in Table 3 in the main text, and we have added a discussion in Appendix Section A.5.2.

---

> > > ### Author Response · Authors · 2020-11-19
> > > **Response to AnonReviewer3 (Part 3)**
> > >
> > > 5. I notice that you cite Wu's article in related works, but there is a lack of comparison to it in later experiment part. In my opinion, the results of the paper and its sequel on time-steps for training SNNs from scratch are worth-noticing. I would like a more comprehensive and fair comparison of results. Wu, Yujie, et al. "Spatio-temporal backpropagation for training high-performance spiking neural networks." Frontiers in neuroscience 12 (2018): 331. Wu, Yujie, et al. "Direct training for spiking neural networks: Faster, larger, better." Proceedings of the AAAI Conference on Artificial Intelligence. Vol. 33. 2019.
> > >
> > > Ans. We thank the reviewer for pointing us to these references. Indeed, the 2 articles mentioned are salient in the context of training SNNs with low inference latency. To compare with Wu et al. (2018), we performed additional experiments implementing the same architecture they have used, having 2 convolution layers (20 channels with 5 × 5 kernels, 30 channels with 5 × 5 kernels, respectively), 2 × 2 average-pooling layers after each convolution layer, followed by 2 fully connected layers (256 and 10 neurons, respectively). The authors in Wu et al. (2018) reported 50.7% accuracy with 30 timesteps, whereas DCT-SNN achieved 68.1% with 28 timesteps on CIFAR-10 dataset.
> > >
> > > Wu et al. (2019) extend their previous work and report 90.53% accuracy on CIFAR-10 in just 12 timesteps using a network with 5 convolutional and 2 fully connected layers. We tried to implement this algorithm ourselves, since the source code was not public. We upload our version of their code along with our supplementary material under the name wu_direct_training.ipynb. However, we could not reproduce the results reported, since the hyperparameters of implementation are not provided in the paper (for eg. learning rate, decay constant and ‘v ‘parameter in Eqn. (9) of the paper), and we did not have enough time to perform grid search to optimize them. Hence, we focus on the critical conceptual differences between their method and ours, listed below.
> > >
> > > Usually in SNNs, the layerwise activations are binary (spikes of 1/0), and hence the synaptic operations are sparse and based on accumulation (Acc) instead of the full precision Multiply-And-Accumulate (MAC) used in ANNs. This is the primary source of energy efficiency of SNNs compared to ANNs, as explained in section 4 of our manuscript. However, in Wu et al. (2019), after each conv layer, the binary activations go through a channel-wise normalization (termed as neunorm in the paper). Essentially, this converts the activations from binary to analog. From Eqns. (9) and (10) in the paper, the operation of normalization involves subtracting the average of the spikes, multiplied with a learnable matrix U from the spikes. This is done before convolving with the convolutional kernel of the next layer. This makes the activations fully analog (floating point) instead of binary spikes, thus reverting the synaptic operations back to MACs instead of Accs, Essentially, this implies that the resulting energy consumption for this case would be at least equivalent to (12*5.1 = 61 timesteps), considering the MAC/Acc energy requirements. Moreover, in SNNs, the layerwise activations are usually very sparse, i.e. only a fraction of total neuron counts are 1’s). However, in Wu et al. (2019), all neurons take part in the MAC operations since the 0’s get shifted during normalization. We argue that all these factors would cause their network to have significantly higher power consumption compared to usual SNNs. The authors do not report energy consumption for their network. Additionally, due the extra feature map at each layer used for normalization, the number of learnable parameters (that make up U) at each layer also increase.  Crucially, we are not clear whether the efficiency of 12 timesteps is arising due to their encoding scheme, because of the proposed channel-wise normalization resulting in analog computation at each layer, or their voting scheme based on class-wise populations. We believe that the analog nature of computation at all the layers makes this network closer to ANNs than SNNs, hence contributing to an extreme reduction in timesteps.
> > >
> > > We have appended both these results to Table 3, and have included this discussion in the main text in Section 4.

---

> > > > ### Author Response · Authors · 2020-11-19
> > > > **Response to AnonReviewer3 (Part 4)**
> > > >
> > > > References
> > > >
> > > > Rueckauer, B., Lungu, I. A., Hu, Y., Pfeiffer, M., & Liu, S. C. (2017). Conversion of continuous-valued deep networks to efficient event-driven networks for image classification. Frontiers in neuroscience, 11, 682.
> > > >
> > > > Lu, S., & Sengupta, A. (2020). Exploring the Connection Between Binary and Spiking Neural Networks. arXiv preprint arXiv:2002.10064.
> > > >
> > > > Wu, Yujie, et al. "Spatio-temporal backpropagation for training high-performance spiking neural networks." Frontiers in neuroscience 12 (2018): 331
> > > >
> > > > Wu, Yujie, et al. "Direct training for spiking neural networks: Faster, larger, better." Proceedings of the AAAI Conference on Artificial Intelligence. Vol. 33. 2019
> > > >
> > > > Rathi, Nitin, et al. "Enabling deep spiking neural networks with hybrid conversion and spike timing dependent backpropagation." arXiv preprint arXiv:2005.01807 (2020)
> > > >
> > > > Park, Seongsik, et al. "T2FSNN: Deep Spiking Neural Networks with Time-to-first-spike Coding." arXiv preprint arXiv:2003.11741 (2020).
> > > >
> > > > P. U. Diehl et al., “Fast-classifying, high-accuracy spiking deep networks through weight and threshold balancing,” in IJCNN, 2015
> > > >
> > > > B. Rueckauer et al., “Conversion of continuous-valued deep networks to efficient event-driven networks for image classification,” Frontiers in Neuroscience, vol. 11, p. 682, 2017.
> > > >
> > > > J. Kim et al., “Deep neural networks with weighted spikes,” Neurocomputing, vol. 311, pp. 373–386, 2018
> > > >
> > > > S. Park et al., “Fast and efficient information transmission with burst spikes in deep spiking neural networks,” in DAC, 2019.

---

### Author Response · Authors · 2020-11-24
**Thank you to the Reviewers**

We would like to thank all the reviewers for their time and suggestions; it has helped us improve our manuscript. The incorporated changes are highlighted in red. Please let us know in case of any more questions or comments!

---

### Decision · Program_Chairs · 2021-01-07
**Final Decision**

**Decision:**

Reject

**Comment:**

This paper provides a method of encoding inputs to a spiking neural network (SNN) using the discrete cosine transform (DCT). The goal is to create a more energy and time efficient means of doing inference with SNNs. The authors provide a description of the method, then show accuracy results on a variety of standard benchmarks. They also compare to a number of other methods for ANN and SNN based inference in the literature. Altogether, they show that their method allows for accurate inference using fewer spikes than other approaches, which can potentially reduce the energy used for inference.

The paper is fairly clearly written, and the results well articulated. The reviewers had a number of concerns, most notably related to questions of (1) clarity about the actual benefits of this approach, and (2) fair comparisons to other models. Altogether, the authors did try to address the reviewers comments, and at least one reviewer increased their score.

However, the actual scores for this paper remained very close to the acceptance threshold, and the first point was difficult to rebut without a lot more added to the paper. Ultimately, this paper is using a classic signal processing strategy to improve SNN run time, and the reviewers asked for some reason as to why that is desirable/novel. The author's answer was effectively that SNNs provide promise for low-energy edge computing, and their method could make SNNs for edge computing even more efficient.  This is potentially of interest for edge computing, but the paper could do a lot more to demonstrate that. Specifically, some consideration of how this would actually operate on spiking chips or a more robust estimate of energy efficiency than that given at the end of section 4 would be required to make this paper a clear accept. Notably, the paper does not demonstrate that this technique could be used to significantly reduce the energy requirements for spiking chips, relative to other SNNs, just that this is more energy efficient than ANNs, which is already known for other spiking neural network approaches. Given this, and the scores relative to other papers at ICLR, a "reject" is recommended. However, the AC notes that this was a difficult decision, and this paper was right at the threshold.